# Temporally specific engagement of distinct neuronal circuits regulating olfactory habituation in *Drosophila*

**Ourania Semelidou[1,2], Summer F Acevedo[1†], Efthimios MC Skoulakis[1]***

[1]Division of Neuroscience, Biomedical Sciences Research Centre "Alexander Fleming", Vari, Greece; [2]School of Medicine, University of Crete, Heraklion, Greece

**Abstract** Habituation is the process that enables salience filtering, precipitating perceptual changes that alter the value of environmental stimuli. To discern the neuronal circuits underlying habituation to brief inconsequential stimuli, we developed a novel olfactory habituation paradigm, identifying two distinct phases of the response that engage distinct neuronal circuits. Responsiveness to the continuous odor stimulus is maintained initially, a phase we term habituation latency and requires Rutabaga Adenylyl-Cyclase-depended neurotransmission from GABAergic Antennal Lobe Interneurons and activation of excitatory Projection Neurons (PNs) and the Mushroom Bodies. In contrast, habituation depends on the inhibitory PNs of the middle Antenno-Cerebral Track, requires inner Antenno-Cerebral Track PN activation and defines a temporally distinct phase. Collectively, our data support the involvement of Lateral Horn excitatory and inhibitory stimulation in habituation. These results provide essential cellular substrates for future analyses of the molecular mechanisms that govern the duration and transition between these distinct temporal habituation phases.

**Editorial note:** This article has been through an editorial process in which the authors decide how to respond to the issues raised during peer review. The Reviewing Editor's assessment is that all the issues have been addressed (see decision letter).

DOI: https://doi.org/10.7554/eLife.39569.001

*For correspondence: skoulakis@fleming.gr

Present address: †Stephens & Associates Contract Research Organization, Texas, United States

Competing interests: The authors declare that no competing interests exist.

## Introduction

Habituation is a highly conserved behavioral modification whereby responses to repetitive or continuous stimuli not associated with concurrent salient stimuli or events are attenuated (*Harris, 1943*). Habituation devalues the salience of a stimulus permitting animals to attend other, potentially more significant stimuli. Importantly, preventing premature habituation is essential to maintain information content long enough to allow association with other stimuli. This led to the notion that habituation is a 'building block for associative learning'.

Habituation paradigms have been used to assess cognitive abilities (*Chard et al., 2014*) and recent studies indicate that genes involved in intellectual disability are linked to impaired habituation (*Lugtenberg et al., 2016*; *Stessman et al., 2016*). Habituation deficiencies have also been linked to disorders, such as schizophrenia (*Akdag et al., 2003*; *Ludewig et al., 2003*), migraines (*Kalita et al., 2014*; *Kropp et al., 2015*), attention-deficit/hyperactivity disorder (*Jansiewicz et al., 2004*; *Massa and O'Desky, 2012*) and autism-spectrum disorders (*Bruno et al., 2014*; *Lovelace et al., 2016*; *Tam et al., 2017*). The implication of habituation in multiple cognitive disorders and its potential effects on associative learning highlight the significance of understanding the molecular mechanisms and neuronal circuitry that govern it.

*Drosophila* is a premier system for molecular approaches to understand habituation because of its advanced molecular and classical genetics. In fact, it is a well-established model for habituation of

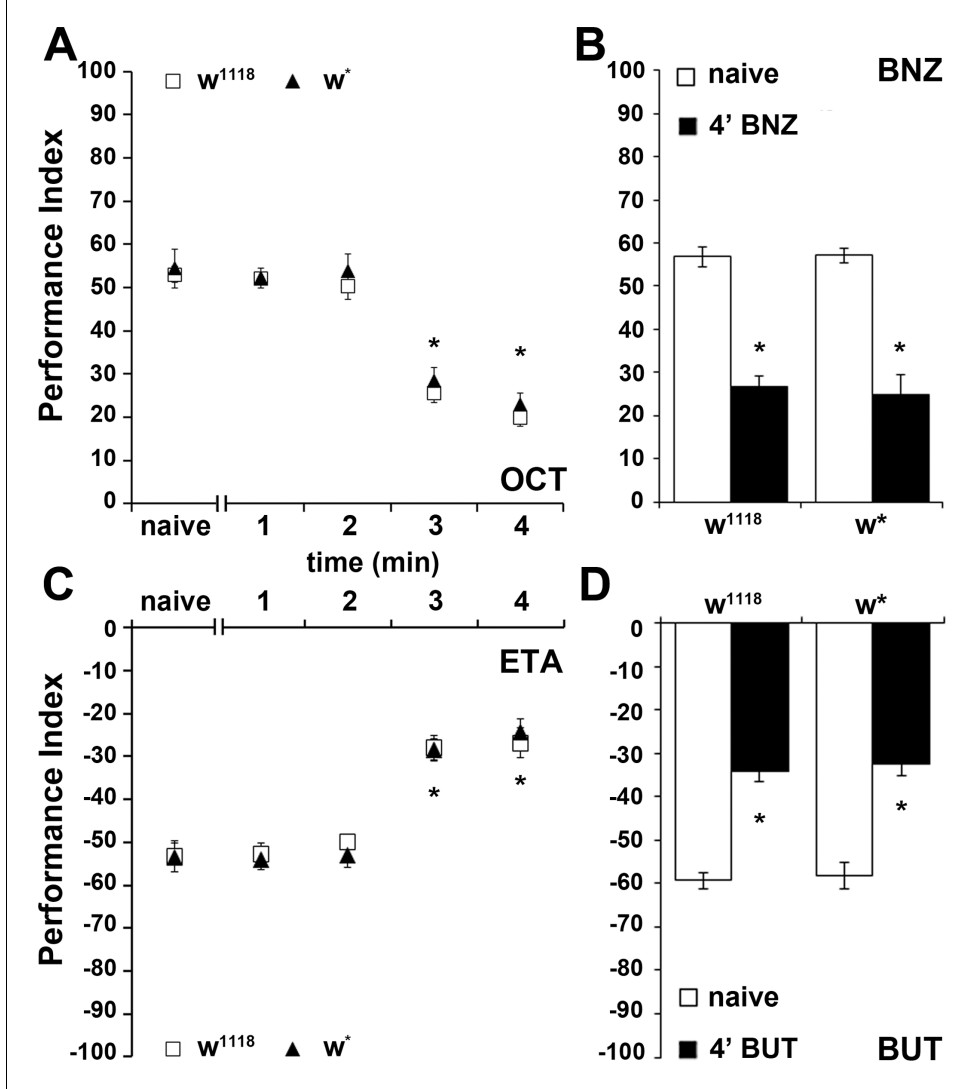

**Figure 1.** An experience-dependent odor-specific decrement in osmotaxis. Mean Performance Indices calculated as detailed in Material and methods are shown ±SEM. Positive values indicate aversion of the odorant and movement towards the air-bearing arm. Negative values indicate attraction to the odorants. Stars indicate significant differences. (A) Pre-exposure of the two control strains (w[1118] and w*) to the aversive odorant 3-Octanol (1X OCT) for 3 or 4 min results in significant avoidance attenuation (p<0.0001, n ≥ 6 for all groups), compared to flies that did not experience the odor except during testing (naive). (B) In contrast to naïve animals, pre-exposure to the aversive odor Benzaldehyde (1X BNZ) for 4 min resulted in significantly attenuated response in both strains (p<0.0001, n ≥ 6 for all groups). (C) Pre-exposure of w[1118] and w* flies to the attractant Ethyl Acetate (1X ETA) for 3 or 4 min precipitated a significant reduction in its attraction (p<0.0001, n ≥ 7 for all groups). (D) Exposure of both control strains to the attractive 2,3-Butanedione (1X BUT) for 4 min decreased significantly its attraction (p<0.0001, n ≥ 7 for all groups). Detailed statistics are found on *Supplementary file 1* and all data are presented in *Figure 1—source data 1*.

DOI: https://doi.org/10.7554/eLife.39569.002

The following source data and figure supplements are available for figure 1:

**Source data 1.** An experience-dependent odor.
DOI: https://doi.org/10.7554/eLife.39569.005

**Figure supplement 1.** Continuous and pulsed OCT stimulation has the same effect on subsequent osmotaxis.
DOI: https://doi.org/10.7554/eLife.39569.003

**Figure supplement 1—source data 1.** Continuous and pulsed OCT stimulation has the same effect on subsequent osmotaxis.
DOI: https://doi.org/10.7554/eLife.39569.004

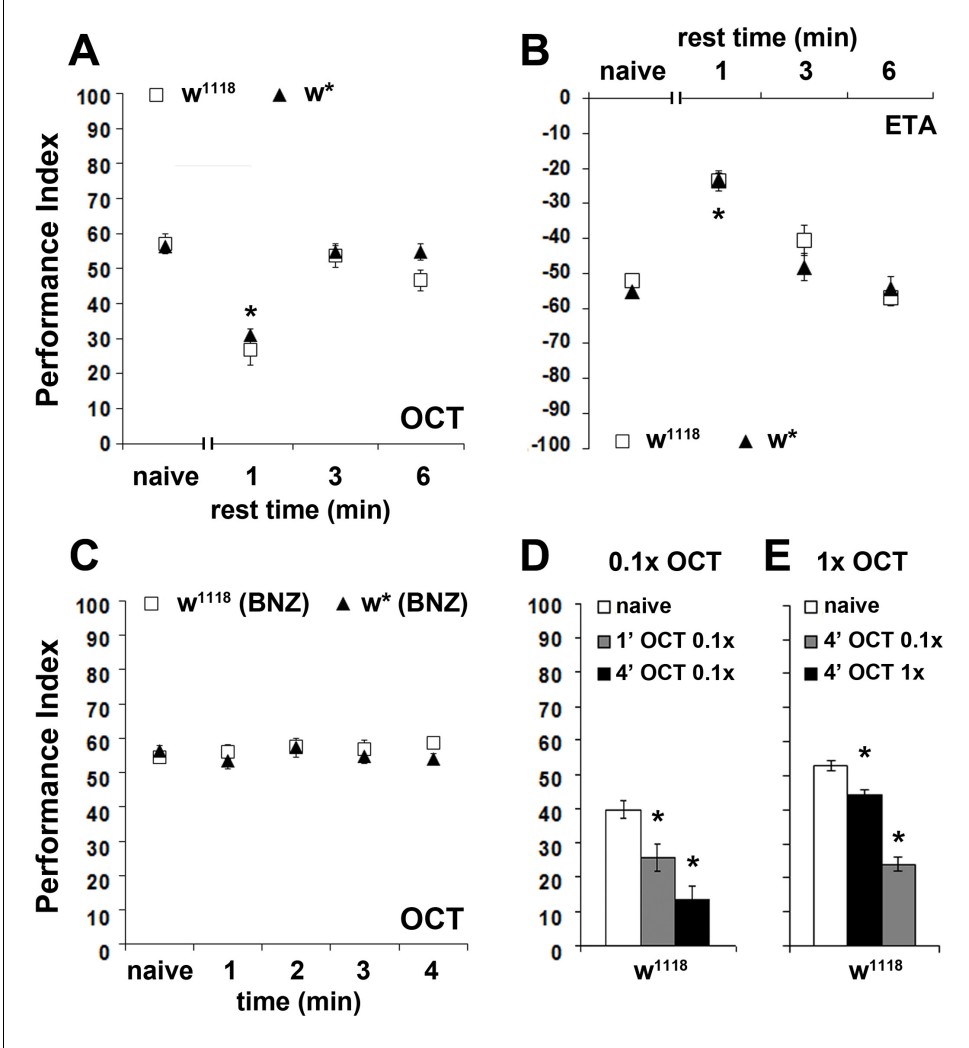

**Figure 2.** The osmotactic response attenuation conforms to habituation parameters. Mean Performance Indices ± SEM are shown in all figures. Positive values indicate aversion, while negative values indicate attraction. Stars indicate significant differences from the naïve response unless specified otherwise. (**A**) A 3 or 6 min rest after a 4-min exposure results in spontaneous recovery of 1X OCT avoidance. Spontaneous recovery was not observed after 1 min of rest (p<0.0001, n ≥ 9 for all groups). (**B**) A 3 or 6 min rest following a 4-min exposure resulted in spontaneous recovery of attraction to 1X ETA, whereas 1 min of rest did not (p<0.0001, n ≥ 7 for all groups). (**C**) Pre-exposure to 1X BNZ for 1–4 min did not result in significant osmotactic decrement in subsequent 1X OCT avoidance (ANOVA p=0.7192, n ≥ 7 for all groups). (**D**) Pre-exposure and testing with 0.1X OCT results a significant decrease in response both after 4 min (black bar, p<0.0001), and after only 1 min of exposure (p=0.0086). (n ≥ 13 for all groups) (**E**) 4 min of exposure to 0.1X OCT followed by testing with 1X OCT (black bar) precipitated significant osmotactic attenuation (p=0.0096), but 4 min of pre-exposure to 1X OCT yielded deeper attenuation (p<0.0001) that was significant different to that of 0.1X OCT (p<0.0001). (n ≥ 8 for all groups) Detailed statistics are found on *Supplementary file 1* and all data are presented in *Figure 2—source data 1*.
DOI: https://doi.org/10.7554/eLife.39569.006

The following source data and figure supplements are available for figure 2:

**Source data 1.** The osmotactic response attenuation conforms to habituation parameters.
DOI: https://doi.org/10.7554/eLife.39569.009
**Figure supplement 1.** Exposure to the attractive odor ETA does affect subsequent OCT avoidance.
DOI: https://doi.org/10.7554/eLife.39569.007
**Figure supplement 1—source data 1.** Exposure to the attractive odor ETA does affect subsequent OCT avoidance.
DOI: https://doi.org/10.7554/eLife.39569.008

various sensory modalities such as taste (*Cevik and Erden, 2012*), vision (*Soibam et al., 2013*), mechanosensory (*Acevedo et al., 2007a*) and escape responses (*Engel and Wu, 2009*), reflecting that habituation is apparent in most, if not all, circuits and modalities of the nervous system. However, in most of these paradigms, the circuits engaged to process the stimulus and establish the experimentally measured attenuated behavioral response are unclear. Importantly, the advanced understanding of the *Drosophila* olfactory circuitry and stimulus processing facilitates exploration of the mechanisms mediating decreased stimulus responsiveness and habituation to inconsequential odors. Such a recently described paradigm of olfactory habituation in *Drosophila* required 30 min of odor exposure and was mediated entirely by antennal lobe neurons (*Das et al., 2011*). In contrast, habituation to repetitive 30 s odor pulses required functional Mushroom Bodies (*Cho et al., 2004*), neurons on the central brain also implicated in associative learning and memory in flies (*Cognigni et al., 2018*).

To resolve this paradox, we focused on the early behavioral dynamics of habituation upon continuous odor stimulation. To that end, we developed and characterized a novel habituation paradigm to rather brief continuous odors. The behavioral responses define two distinct phases, an initial phase we term habituation latency, when stimulus responsiveness is maintained, which is followed by a significant response decrement reflecting habituation. Analogous response dynamics have been reported for footshock habituation (*Acevedo et al., 2007a*). In addition, we investigated whether these phases engage and are mediated by distinct neuronal circuits. The results highlight the stimulus duration-dependent activation of specific neuronal subsets and their distinct roles in securing timely habituation latency and habituation induction.

## Results

### An experience-dependent odor-specific decrement in osmotaxis

We used continuous exposure to odorants adjusted to elicit relatively mild aversive (3-octanol-OCT and benzaldehyde-BNZ) and attractive (ethyl acetate-ETA and 2,3-butanedione-BUT) osmotactic responses (*Acevedo et al., 2007b*) (*Figure 1A–D*). After 4 min of exposure to OCT, a highly significant ~60% avoidance attenuation was presented by both $w^{1118}$ and $w^*$ controls (*Figure 1A*). Similarly, a 50% decrease in BNZ avoidance was apparent after a 4-min exposure (*Figure 1B*). Moreover, attraction to ETA (*Figure 1C*) and BUT (*Figure 1D*) were similarly abated after a 4-min exposure to the respective odorants, suggesting an experience-dependent decrease in osmotaxis. Failure to avoid or move toward the test stimulus could not be attributed to odor-induced locomotor impairments, because most flies left the choice point in the absence of test odors and both naive and pre-exposed animals distributed equally in the arms of the maze (not shown). Because the osmotactic attenuation was similar irrespective of odor valence, we used the milder aversive OCT for all subsequent experiments. Intriguingly, the initial 120 s of exposure define an osmotactic attenuation latency period with the odor apparently retaining its value and the flies responding as if naive (*Figure 1*). Latency to habituate with similar dynamics has also been described in the footshock habituation paradigm (*Acevedo et al., 2007a*) and appears operant in other *Drosophila* habituation paradigms that examined this early phase such as for the electrically induced giant fiber response (*Engel and Wu, 2009*).

Is the osmotactic attenuation consequent of continuous exposure per se, or of the total stimulus exposure time irrespective of delivery method? To address this, the flies were subjected to discrete OCT pulses totaling the same exposure time as upon continuous exposure (see Materials and methods). The interstimulus interval (ITI) was kept around 25% of each pulse, because adaptation, which we aimed to avoid, has been reported proportional to odor stimulus duration (*de Bruyne et al., 1999*). Significantly, OCT avoidance remained at naive levels after a single 1 min or two 30 s (with 8 s ITI) exposures (*Figure 1—figure supplement 1*). Moreover, avoidance was equally attenuated by one continuous 4 min exposure or 4, 1 min OCT pulses (15 s ITI) (*Figure 1—figure supplement 1*). These results indicate that attenuation of the avoidance response depends on total time of odor exposure, but not the mode of its delivery.

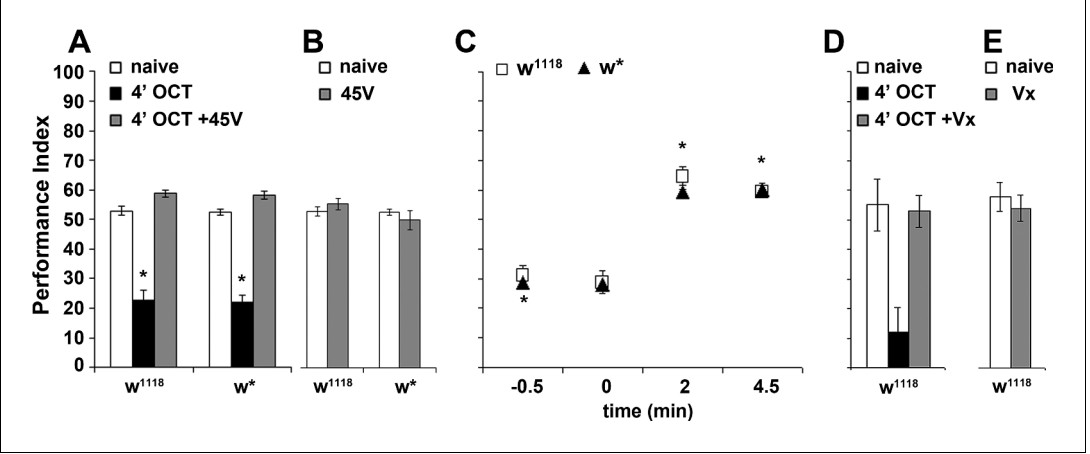

**Figure 3.** Dishabituation with mechanosensory stimuli results in recovery of the naive response. Mean Performance Indices ± SEM are shown in all figures. Stars indicate significant differences from the naive response unless specified otherwise. (A) Application of a 45V electric footshock after the 4 min odor exposure (grey bar) leads to reversal of the habituated response (black bar) in both $w^{1118}$ and w* control strains (p=0.1412 and 0.0873, n $\geq$ 8 for all groups). (B) Application of a 45V electric footshock to naïve animals does not affect their response to the odorant (ANOVA p=0.3461, n $\geq$ 8 for all groups). (C) A 45V electric footshock was applied 30 s before (−0.5), concurrent with the onset of odor exposure (0), 2 min into the odor exposure (2), or 30 s post-exposure prior to testing. Dishabituation was evident only when the shock was delivered during or just after 1X OCT exposure (p<0.0001 for 2 and 4.5 min compared to −0.5, n $\geq$ 7 for all groups). (D) Application of a 3 s vortex at maximum speed (grey bar) after the habituating 4-min odor exposure (black bar, p=0.0010) led to recovery (p=0.9729) of the naïve response (open bar). (n = 10 for all groups) (E) Application of a 3 s vortex to naive animals did not affect their response to the odorant (ANOVA p=0.5460, n = 18 for all groups). Detailed statistics are found on *Supplementary file 1* and all data are presented in *Figure 3—source data 1*.

DOI: https://doi.org/10.7554/eLife.39569.010

The following source data is available for figure 3:

**Source data 1.** Dishabituation with mechanosensory stimuli results in recovery of the naive response.
DOI: https://doi.org/10.7554/eLife.39569.011

## The decrement in osmotactic response conforms to habituation parameters

Because it requires 4 min of total odor exposure to precipitate an osmotactic decrement, we wondered whether this behavioral response conforms to the classically defined habituation parameters of Thompson and Spencer (*Thompson and Spencer, 1966*; *Rankin et al., 2009*). Accordingly, animals that habituate to a repetitive or continuous stimulus should spontaneously recover if the stimulus is withheld. Indeed, a 3 or 6 min post-exposure rest resulted in spontaneous recovery of the osmotactic response to naive levels, both for aversive and attractive odors (*Figure 2A,B*).

The habituated response often exhibits generalization after exposure to similar stimuli (*Thompson and Spencer, 1966*; *Rankin et al., 2009*). Therefore, we investigated whether the osmotactic attenuation is specific to the pre-exposed odor, or flies generalize, presenting reduced responses to different odors, but of similar valence. After a habituation-inducing 4-min exposure to BNZ, flies avoided OCT normally (*Figure 2C*), suggesting no generalization of the response decrement to a different aversive odor. Moreover, pre-exposure to odors of opposite valence, such as the attractive ETA, also did not alter subsequent OCT avoidance (*Figure 2—figure supplement 1*), indicating that the osmotactic attenuation is odor-specific, a result also inconsistent with broad sensory fatigue. However, we cannot rule out generalization to similar odors (for example 1-OCT and 3-OCT), which might activate overlapping neurons in circuits necessary for habituation, as previously shown for odorants with similar molecular features in the leg movement habituation paradigm (*Chandra and Singh, 2005*). Given that we have used a limited set of odorants to test generalization, it is difficult to ascertain that osmotactic attenuation is not generalized.

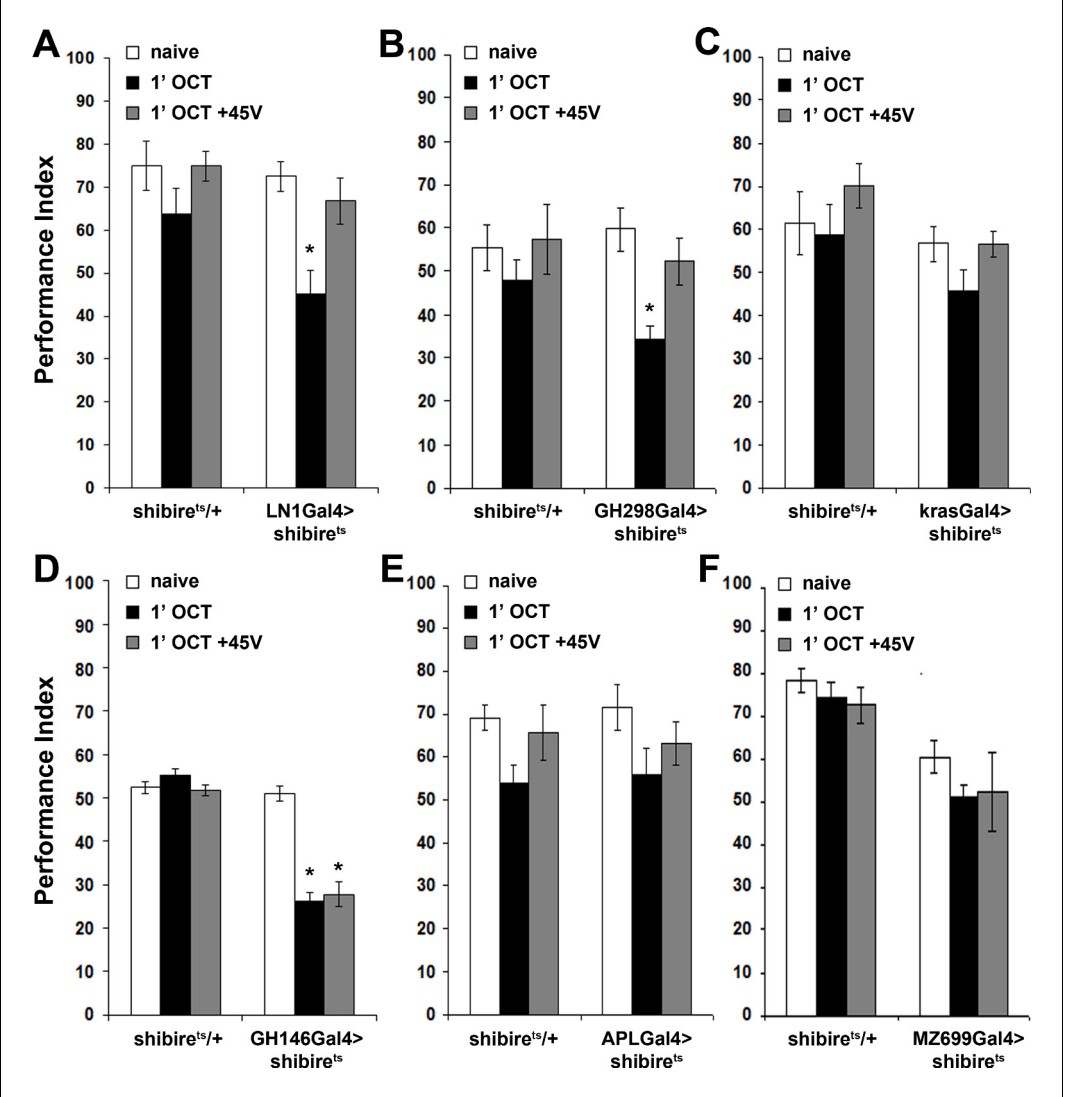

**Figure 4.** Inhibitory local interneurons and excitatory projection neurons are necessary for habituation latency. Mean Performance Indices ± SEM are shown in all figures. Stars indicate significant differences from the naive response unless specified otherwise. (A) Functional silencing of iLNs with UAS-shibire[ts] under LN1GAL4 resulted in premature/facilitated habituation after only 1 min of 1X OCT exposure (black bar, p=0.0012), which recovers to naive levels after a single 45V electric footshock (grey bar, p=0.6268). In contrast, UAS-shibire[ts]/+controls did not present significant differences among treatment groups. (n ≥ 9 for all groups) (B) Functional silencing of the GH298GAL4-marked iLNs with UAS-shibire[ts] facilitates habituation, apparent after 1 min of odor exposure (black bar, p<0.0001), which recovered to naïve levels by a 45V electric footshock (grey bar, p=0.4332). In contrast, UAS-shibire[ts]/+controls did not present significant differences among treatment groups. (n ≥ 10 for all groups) (C) Functional silencing of the krasGAL4-marked eLNs driving UAS-shibire[ts] did not affect responsiveness to 1X OCT after 1 min of exposure. Controls were similarly unaffected (black bar, ANOVA p=0.1171, n ≥ 10 for all groups). (D) Functional silencing of the GH146Gal4-marked excitatory projection neurons with UAS-shibire[ts] resulted in facilitated habituation after 1 min of exposure (black bar, p<0.0001) and blocked footshock dishabituation (grey bar, p<0.0001). Control UAS-shibire[ts]/+flies did not present significant response differences irrespective of treatment. (n ≥ 8 for all groups) (E) Blocked neurotransmission from APL neurons did not affect the response to 1X OCT following 1 min of exposure, similar to the responses of control groups (black bar, ANOVA p=0.1573, n ≥ 11 for all groups). (F) Expression of UAS-shibire[ts] in MZ699GAL4 neurons to functionally silence the iPNs did not precipitate differences from the naïve response, or from control flies (black bar, ANOVA p=0.4033, n ≥ 10 for all groups). Detailed statistics on *Supplementary File 1* and all data are presented in *Figure 4—source data 1*.
DOI: https://doi.org/10.7554/eLife.39569.012

The following source data and figure supplements are available for figure 4:

*Figure 4 continued on next page*

*Figure 4 continued*

**Source data 1.** Inhibitory local interneuronsand excitatoryprojection neuronsare necessary for habituation latency.
DOI: https://doi.org/10.7554/eLife.39569.017

**Figure supplement 1.** Expression of Tetanus Toxin Light Chain (TTX) under GH146Gal4 results in premature habituation
DOI: https://doi.org/10.7554/eLife.39569.013

**Figure supplement 1—source data 1.** Expression ofTetanus Toxin Light Chain(TTX) under GH146Gal4 results in premature habituation.
DOI: https://doi.org/10.7554/eLife.39569.015

**Figure supplement 2.** GABA attenuation in GH146-marked neurons does not affect habituation latency.
DOI: https://doi.org/10.7554/eLife.39569.014

**Figure supplement 2—source data 1.** GABA attenuation in GH146-marked neurons does not affect habituation latency.
DOI: https://doi.org/10.7554/eLife.39569.016

Because weaker stimuli induce faster or more pronounced habituation (*Rankin et al., 2009*), we investigated the effect of stimulus strength by exposing flies to dilute 0.1X OCT for 1 or 4 min and testing against 0.1X OCT. In agreement with *Rankin et al., 2009*, the diluted odor attenuated the response after only 1 min (*Figure 2D*), instead of the 4 min required for 1X OCT (*Figure 1A*), indicating that the weaker stimulus shortened the osmotactic attenuation latency as expected. Moreover, this response depended on the strength of the testing stimulus, because exposure to 0.1X OCT and testing against 1X OCT resulted in significant attenuation after 4 min, albeit decreased in magnitude compared to the decrement after exposure to 1X OCT (*Figure 2E*). Although the firing frequency of the OCT-responsive OSNs to different odorant concentrations is not known, imaging experiments on glomerulus activation indicated concentration dependent activation, with high concentration doubling the glomerular response (*Yu et al., 2004*). Combined with the faster habituation upon exposure to lower OCT concentration, this suggests that the concentrations used are likely interpreted as different stimulus strengths.

Moreover, these results confirm that the response decrement is not a consequence of OSN adaptation, which occurs after odor pulses even as brief as less than 30 s and dynamically adjusts odor sensitivity (*Cao et al., 2016*). If the decrement depended on OSN adaptation, increased odor concentration would lead to faster adaptation and thus, faster response attenuation, but the opposite was observed. In addition, our data indicate that 1-min OCT exposure does not affect subsequent response to the same stimulus, strongly suggesting that it is not the OSNs, but neurons downstream in the olfactory pathway, that are implicated in the response decrement.

The experience-dependent osmotactic attenuation can be formally considered habituation if a relatively strong unrelated stimulus restores the naive response (dishabituation) (*Thompson and Spencer, 1966*; *Rankin et al., 2009*). We attempted dishabituation using two distinct mechanical stimuli, electric footshock and vortexing. The strength and number of footshock stimuli required to reverse the osmotactic decrement with the weakest possible footshock were determined experimentally (see Materials and methods). A single 45-Volt footshock delivered after the 4-min OCT exposure restored subsequent avoidance to naive levels (*Figure 3A*) but did not affect OCT avoidance in naive flies (*Figure 3B*). The dishabituator should not be effective prior to odor exposure. Indeed, the footshock was effective only if delivered after OCT exposure, at the end of the apparent ~120 s latency period, but not prior to, or at the onset of odor presentation (*Figure 3C*). This suggests that the footshock likely interferes with processes occurring and potentially mediating habituation onset, as if effectively re-setting the latency period. Moreover, 3 s of vortexing immediately after the 4-min OCT exposure also resulted in recovery of the naïve response (*Figure 3D*), but did not affect responsiveness of vortexed but naive flies to OCT (*Figure 3E*).

Recovery of the response after footshock or vortexing and conformation to all other examined parameters demonstrates that the experience-dependent response attenuation is in fact olfactory habituation. Moreover, the initial ~120 s period represents habituation latency, potentially facilitating stimulus salience evaluation as previously suggested for habituation to repetitive electric footshocks (*Acevedo et al., 2007a*). If so, shortening or eliminating the latency period would result in early devaluation of the stimulus, effectively premature habituation.

## Inhibitory local interneurons are necessary for habituation latency

Olfactory information is conveyed through the olfactory sensory neurons (OSNs) to the antennal lobe, consisting of local interneurons (LNs) and projection neurons (PNs) that transmit the information to higher order structures (*Masse et al., 2009*). Identification of neuronal subsets engaged in the latency period and habituation to the aversive OCT was facilitated by the well-defined olfactory circuitry in *Drosophila*.

We initially focused on the antennal lobe LNs, which are mostly (95%) GABAergic, activated both by sensory and projection neurons, and modulate PN output (*Silbering et al., 2008*; *Tanaka et al., 2012*). Potentiation of GABAergic inhibition from the LNs onto PNs has been suggested to mediate habituation after 30 min of odor exposure (*Das et al., 2011*). To determine whether LNs function similarly upon 4-min exposure, we conditionally blocked their synaptic output by transgenically expressing therein the temperature sensitive dynamin Shibire[ts] (Shi[ts]). At the restrictive temperature Shi[ts] adopts an inactive conformation, blocking neurotransmitter reuptake, thus silencing neurons by depletion of the releasable neurotransmitter pool (*Kitamoto, 2001*).

Shi[ts] was expressed under the LN1Gal4 and GH298Gal4 drivers, which mark antennal lobe GABAergic inhibitory local interneurons (iLNs) (*Acebes et al., 2011*; *Okada et al., 2009*), arborizing in most glomeruli (*Tanaka et al., 2012*) and presenting extensive contacts with the PNs (*Tanaka et al., 2009*). We asked whether silencing iLNs altered habituation latency by exposing the flies to OCT for only 1 min. Interestingly, 1 min exposure attenuated OCT avoidance, whereas controls retained their naive response (*Figure 4A,B*). The attenuated osmotaxis is in fact premature habituation demonstrated by its reversal to naive levels by a single post-pre-exposure dishabituating footshock (*Figure 4A,B*). It should be noted that for simplicity all driver heterozygotes are not presented in *Figure 4*, but their performance (Suppl File 2), was similar to the *shibire[ts]*/+controls. Therefore, antennal lobe GABAergic iLNs appear to have a dual role. Upon brief odor exposure, they modulate antennal lobe activity to preserve stimulus value, contributing to habituation latency, while upon prolonged, 30-min exposure they were reported to facilitate habituation (*Das et al., 2011*).

In contrast, silencing the mostly (60–66%) cholinergic, excitatory LNs (eLNs) under krasavietzGal4 (*Acebes et al., 2011*; *Shang et al., 2007*) did not yield significant effects (*Figure 4C*). However, eLNs are also connected to ePNs and to iLNs via electrical synapses. These electrical synapses modulate PN output through direct excitation upon exposure to weak stimuli and indirect inhibition after strong stimulus exposure (*Yaksi and Wilson, 2010*; *Huang et al., 2010*). Although the chemical component of these eLN synapses is not implicated and as Shi[ts] does not affect gap junction activity, we cannot rule out the possibility that electrical coupling of these synapses contributes to maintenance of the avoidance response, by maintaining excitation.

Collectively then, modulation of PN output by inhibitory GABAergic neurotransmission from the iLNs, and possibly via eLN electrical synapses, is crucial for stimulus value preservation and inhibition of premature habituation. LNs are known to modulate the PN output by broadening or narrowing their response dynamics (*Silbering et al., 2008*). LN activation also mediates periodic PN inhibition, thought to synchronize the responses of the latter and impact activation of Mushroom Body neurons (*MacLeod and Laurent, 1996*). Furthermore, iLN activation decreases PN firing rates at high odor concentrations (*Wilson and Laurent, 2005*). Hence, attenuating iLN output may functionally mimic exposure to low odor concentration, which results in faster habituation (*Figure 2D*). Alternatively, signals un-modulated by the iLNs could be interpreted as 'noise', decreasing stimulus salience and facilitating habituation. In contrast, eLNs function in gain and redistribution of odor-evoked activity over a larger ensemble of PNs at low odor concentrations (*Shang et al., 2007*). Since we use a relatively strong odor, we cannot exclude a role for the eLNs in habituation to dilute odorants. In addition, since strong stimulus exposure could lead to iLN activation via eLN-iLN gap junctions (*Yaksi and Wilson, 2010*), it is possible that these chemical synapses are also necessary for habituation latency.

## Excitatory projection neurons are essential for habituation latency

Because iLNs modulate PN activity, we examined the role of the latter in habituation latency. PNs form three antennocerebral tracks (inner-iACT, middle-mACT and outer-oACT) (*Stocker, 2001*), connecting the antennal lobe with higher order structures - the Mushroom Bodies (MBs) and the Lateral Horn (LH) (*Tanaka et al., 2004*; *Wong et al., 2002*). Because a driver clearly marking oACT neurons

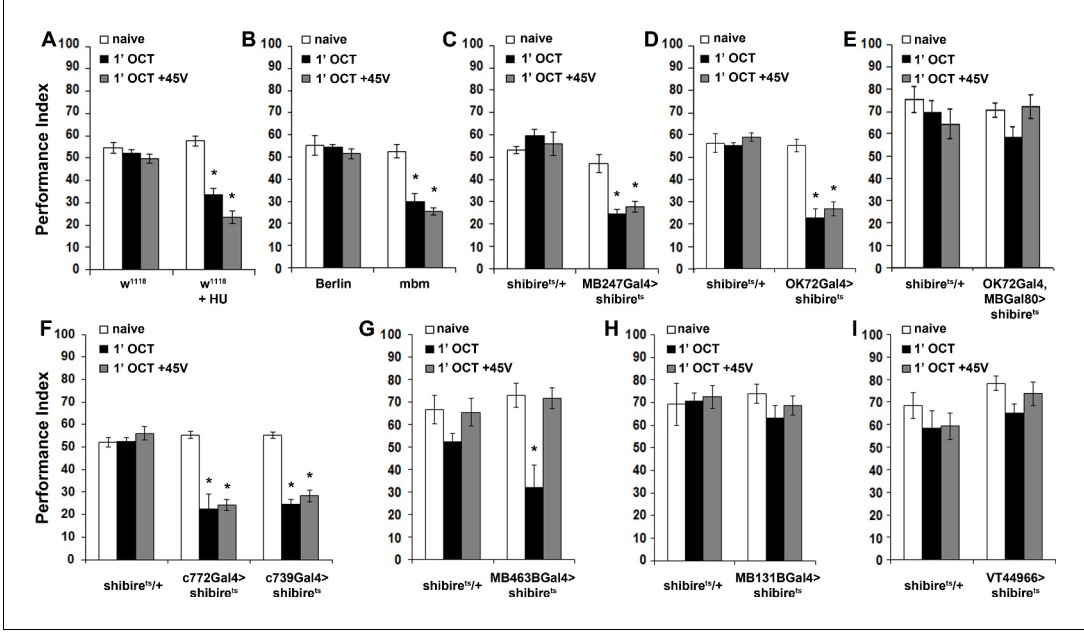

**Figure 5.** The mushroom bodies are essential for habituation latency and normal dishabituation. Mean Performance Indices ± SEM are shown in all figures. Stars indicate significant differences from the naïve response unless specified otherwise. (A) Flies with HU-ablated mushroom bodies (MBs) exhibit premature habituation after a 1-min exposure to 1X OCT (black bar, p<0.0001) and inability to dishabituate (grey bar, p<0.0001), while the response of the control groups was not affected. (n ≥ 8 for all groups) (B) $mbm^1$ mutants also presented facilitated habituation after a 1-min exposure to this odorant (black bar, p<0.0001) and inability to dishabituate (grey bar, p<0.0001). (n ≥ 6 for all groups) (C) Silenced neurotransmission from the MBs with MB247Gal4 driving UAS-shibire[ts] resulted in premature habituation following a 1 min 1X OCT exposure (black bar, p<0.0001) and inability to dishabituate (grey bar, p<0.0001). (n ≥ 8 for all groups) (D) Blocked neurotransmission in OK72Gal4-expressing neurons also facilitated habituation after a 1-min odor exposure (black bar, p<0.0001), and inability to dishabituate (grey bar, p<0.0001). (n ≥ 6 for all groups) (E) Functional silencing of all OK72Gal4-marked neurons apart from the MBs under OK72Gal4;MBGal80 does not affect odor habituation after 1 min of exposure (black bar, ANOVA p=0.0752, n ≥ 10 for all groups). (F) Functional silencing of the αβ MB neurons under c772Gal4 and c739Gal4 facilitated habituation after only 1 min of 1X OCT exposure (black bars, p=0.0002 for c772Gal4 and p<0.0001 for c739Gal4) and resulted in inability to dishabituate (grey bars, p=0.0005 for c772Gal4 and p<0.0001 for c739Gal4) in contrast to controls (UAS-shibire[ts]/+) that did not alter their response. (n ≥ 6 for all groups) (G) Functional silencing of the α'β' MB neurons under MB463BGal4 facilitated habituation after 1 min of 1X OCT exposure (black bar, p=0.0011) in contrast to controls that did not alter their response, and showed normal dishabituation (grey bar, p=0.9842). (n ≥ 8 for all groups) (H-I) Blocked neurotransmission from the γ MB neurons results in normal OCT responsiveness after 1 min of exposure under both MB131BGal4 (black bar, ANOVA p=0.3026, n ≥ 8 for all groups) and VT44966 (black bar, ANOVA p=0.0651, n ≥ 8 for all groups). Detailed statistics on *Supplementary file 1* and all data are presented in *Figure 5—source data 1*.

DOI: https://doi.org/10.7554/eLife.39569.018

The following source data and figure supplements are available for figure 5:

**Source data 1.** Themushroom bodiesare essential for habituation latency and normal dishabituation.

DOI: https://doi.org/10.7554/eLife.39569.023

**Figure supplement 1.** Attraction to 1X ETA was not altered by Hydroxyurea (HU)-dependent ablation of the MBs.

DOI: https://doi.org/10.7554/eLife.39569.019

**Figure supplement 1—source data 1.** Attraction to 1X ETA was not altered by Hydroxyurea (HU)-dependent ablation of the MBs.

DOI: https://doi.org/10.7554/eLife.39569.021

**Figure supplement 2.** Expression of Tetanus Toxin Light Chain (TTX) in the MBs results in premature habituation.

DOI: https://doi.org/10.7554/eLife.39569.020

**Figure supplement 2—source data 1.** Expression ofTetanus Toxin Light Chain(TTX) in the MBs results in premature habituation.

DOI: https://doi.org/10.7554/eLife.39569.022

is not currently available to the best of our knowledge, we used the GH146Gal4 and MZ699Gal4 drivers to silence the iACT and mACT, respectively. GH146Gal4 marks 60% of the PNs, which account for approximately 90 mainly excitatory neurons (*Liang et al., 2013*) that project axons through the iACT and innervate both MBs and LH (*Wong et al., 2002*).

Surprisingly, silencing GH146 neurons did not eliminate OCT avoidance (*Figure 4D*). This suggests that residual PNs not marked by GH146Gal4 or electrical synapses not affected by Shi^ts^-mediated silencing suffice to convey odor information leading to odor avoidance. However, silencing GH146 neurons decreased habituation latency, resulting in premature habituation after only 1 min of OCT exposure and interestingly, eliminated dishabituation (*Figure 4D*). This suggests that neurotransmission from these neurons prevents premature habituation and promotes dishabituation. Independent validation of the reduced habituation latency was obtained by constitutive silencing of GH146 neurons with Tetanus Toxin Light Chain (*Figure 4—figure supplement 1*).

Importantly, besides excitatory neurons giving rise to the iACT, GH146Gal4 also marks the GABAergic and octopaminergic (*Liu and Davis, 2009*; *Wu et al., 2013*) anterior paired lateral (APL) neurons, also known to be activated by odors (*Silbering et al., 2008*). APL neurons innervate the MBs and contribute to associative learning and memory (*Liu and Davis, 2009*; *Wu et al., 2013*). They were silenced under APLGal4 (*Wu et al., 2013*), to determine whether they contribute to the decreased habituation latency upon synaptic blocking under the GH146Gal4 driver. Abrogating APL neurotransmission did not facilitate habituation (*Figure 4E*) and reducing GABA production in GH146Gal4-marked neurons via a GAD-RNAi, also did not shorten habituation latency (*Figure 4—figure supplement 2*). Because the latter is expected to affect both the inhibitory PNs and the GABAergic APL neurons marked by the GH146Gal4 driver, these results confirm that only the excitatory iACT PNs are essential for maintenance of habituation latency.

Furthermore, silencing the 38 mainly GABAergic mACT projection neurons innervating the lateral horn (*Liang et al., 2013*) under MZ699Gal4 (*Tanaka et al., 2009*), did not facilitate habituation after 1 min OCT exposure (*Figure 4F*). This strongly indicates that the mACT is dispensable for habituation latency. However, since Shi^ts^ expression affects only the chemical synapses between the iPNs and the LH, and not gap junctions between iPNs and ePNs (*Shimizu and Stopfer, 2017*), it is possible that iPN-ePN electrical coupling drives maintenance of the response. Electrical coupling of iPNs and ePNs was been shown to amplify the antennal lobe response to certain odorants and may in fact enhance the response to the training odor, thus inhibiting habituation. Together with the GH146Gal4 silencing experiments, these results indicate that innervation of the MBs and LH by excitatory iACT neurons, but not the mACT PNs, is essential to sustain habituation latency and prevent premature habituation to brief continuous odor exposure.

## The MBs are essential for habituation latency and dishabituation

The role of the iACT in response maintenance suggested that the MBs may also be engaged in the latency phase of odor habituation as they are for habituation to ethanol pulses (*Cho et al., 2004*) and electric footshock (*Acevedo et al., 2007a*). If necessary for habituation latency, then eliminating the MBs should lead to premature habituation. To address this hypothesis, we used two complementary approaches. Chemical ablation with hydroxyurea (HU) (*Acevedo et al., 2007a*) and the mbm^1^ mutant presenting structurally aberrant, greatly reduced MBs (*Raabe et al., 2004*). The Berlin strain was used as the appropriate cognate genetic control for mbm^1^ (*Raabe et al., 2004*). Critically, both MB-ablated and mbm^1^ flies avoid OCT and other odorants normally (*Acevedo et al., 2007b*; *Raabe et al., 2004*), verifying in essence that the LH without MB inputs is fully capable of mediating innate, unmodulated avoidance (*Figure 5A,B*-naïve) and attraction (*Figure 5—figure supplement 1*).

MB ablation (*Figure 5A*) or reduction (mbm^1^), led to premature habituation after 1 min of OCT exposure (*Figure 5B*). This drastically reduced habituation latency indicates that normally the MBs inhibit olfactory habituation, possibly by modulating the LH-mediated innate OCT avoidance. To verify this and determine whether the premature habituation of MB-ablated and mbm^1^ flies was developmental in origin, we silenced the MBs using MB247Gal4 (*Zars et al., 2000*). Although driver heterozygotes are not included in *Figure 5*, no significant differences from shibire^ts^/+ were uncovered (*Supplementary file 2*).

Remarkably, 1 min of OCT exposure was sufficient to elicit habituation upon functionally silencing the MBs (*Figure 5C*), just as for MB-ablated and mbm^1^ animals (*Figure 5A,B*). This was

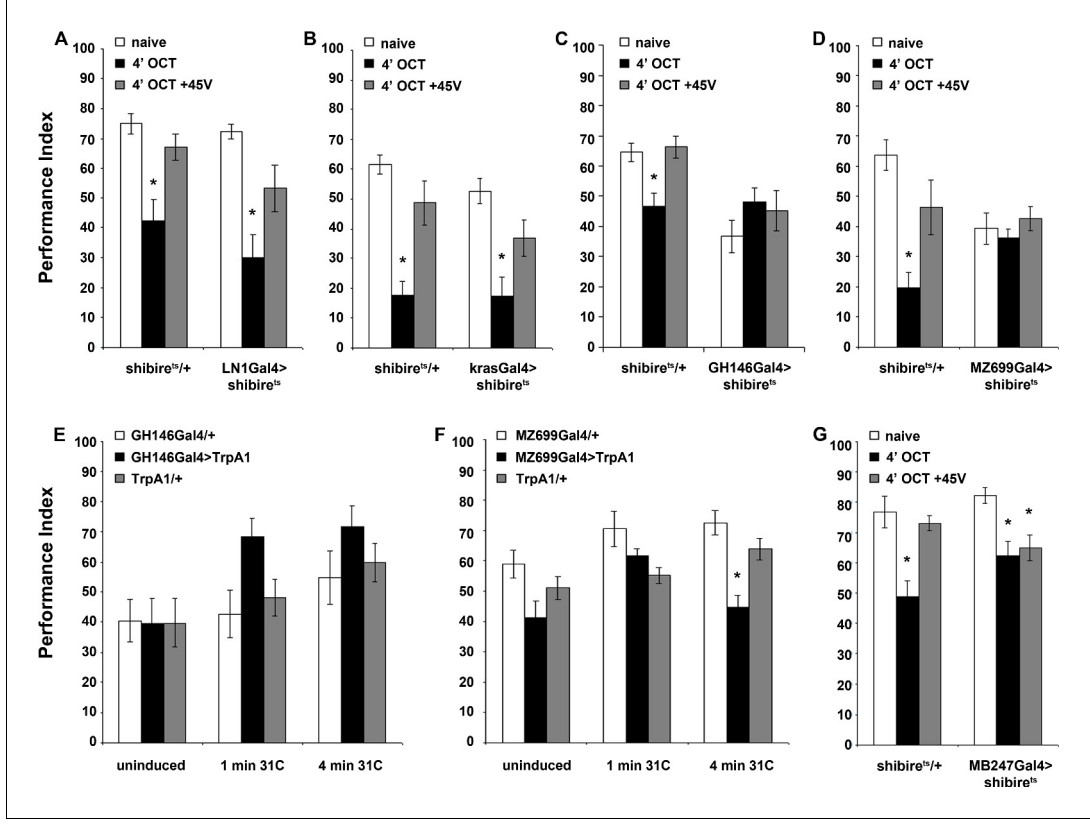

**Figure 6.** Distinct neuronal subsets are required for olfactory habituation. Mean Performance Indices ± SEM are shown in all figures. Stars indicate significant differences from the naive response unless specified otherwise. (**A**) Blocked neurotransmission from iLNs results in normal habituation after 4 min of OCT exposure (black bar, p<0.0001) and normal dishabituation (grey bar, p=0.0867), similar to controls. (n ≥ 10 for all groups) (**B**) Blocked neurotransmission from the eLNs yielded normal habituation to 4 min 1X OCT exposure (black bar, p<0.0001) and normal dishabituation (grey bar, p=0.0977), similar to controls. (n ≥ 12 for all groups) (**C**) Blocked neurotransmission from the GH146Gal4-marked ePNs resulted in abrogated habituation, with the response to OCT remaining at naïve levels even after 4 min of exposure (black bar, ANOVA p=0.3640), while controls habituated (black bar, p=0.0026) and dishabituated normally (grey bar, p=0.9159). (n ≥ 11 for all groups) (**D**) Blocked neurotransmission from iPNs under MZ699Gal4 resulted in abrogated habituation, with the response to 1X OCT remaining at naïve levels after 4 min of exposure (black bar, ANOVA p=0.6121), while controls habituated (black bar, p=0.0002) and dishabituated normally (grey bar, p=0.1491). (n ≥ 10 for all groups) (**E**) Activation of the GH146Gal4-marked PNs with UAS-TRPA1 but without odor exposure did not alter significantly the response to 1X OCT after 1 min (black bar, ANOVA p=0.03) or 4 min of activation (black bar, ANOVA p=0.2707). (n ≥ 10 for all groups) (**F**) Activation of the iPNs under MZ699Gal4 to drive UAS-TRPA1 did not alter significantly the response to 1X OCT after 1 min (black bar, ANOVA p=0.0190), but 4 min of activation sufficed to produce significant habituation (black bar, p<0.0001 when compared to MZ699Gal4/+). (n ≥ 13 for all groups) (**G**) Blocked neurotransmission from the MBs under MB247Gal4 resulted in normal habituation to 4 min of 1X OCT exposure (black bar, p=0.0033) and inability to dishabituate (grey bar, p=0.0075) while control flies habituate (black bar, p=0.0003) and dishabituate normally (grey bar, p=0.8229). (n ≥ 10 for all groups) Statistical details on *Supplementary file 1* and all data are presented in *Figure 6—source data 1*.
DOI: https://doi.org/10.7554/eLife.39569.024

The following source data and figure supplements are available for figure 6:

**Source data 1.** Distinct neuronal subsets are required for olfactory habituation.
DOI: https://doi.org/10.7554/eLife.39569.027

**Figure supplement 1.** GABA attenuation in GH146-marked neurons does not affect habituation.
DOI: https://doi.org/10.7554/eLife.39569.025

**Figure supplement 1—source data 1.** GABA attenuation in GH146-marked neurons does not affect habituation.
DOI: https://doi.org/10.7554/eLife.39569.026

independently validated by expression of Tetanus Light Chain under MB247Gal4 (*Figure 5—figure supplement 2*). Further confirmation for the role of the MBs in habituation latency was obtained with OK72Gal4 (*de Haro et al., 2010*) (*Figure 5D*). Because in addition to the MBs OK72Gal4 marks antennal lobe neurons (*Devaud et al., 2003*), we used OK72Gal4;MBGal80 to silence only the latter but spare the MBs, a manipulation which did not yield facilitated habituation, verifying that the MB component of OK72Gal4 is necessary for habituation latency (*Figure 5E*). Therefore, the MBs are essential for response maintenance during the first minute(s) of odor exposure. In addition, this demonstrates that the facilitated habituation of HU-ablated and *mbm*[1] flies is unlikely consequent of altered development or brain anatomy re-arrangement. Furthermore, structurally and functionally intact MBs are also required for dishabituation, as their complete ablation, partial abrogation in *mbm*[1] mutants, or functional silencing, eliminated dishabituation after 1-min OCT exposure (*Figure 5A–D*).

The three broad types of MB intrinsic neurons, the αβ, α′ β′and γ (*Crittenden et al., 1998*) are differentially implicated in olfactory learning and memory (*Zhang and Roman, 2013*; *Krashes et al., 2007*; *Yu et al., 2006*; *Blum et al., 2009*). Because neurotransmission from αβ neurons prevents premature habituation to footshocks (*Acevedo et al., 2007a*), we investigated whether these neurons function similarly for olfactory habituation. The role of the αβ neurons was examined using c772Gal4 and c739Gal4, which label preferentially αβ and γ neurons and almost exclusively αβ neurons, respectively (*Aso et al., 2009*). Silencing the MBs with either driver resulted in premature habituation after 1 min of OCT exposure and inability to dishabituate (*Figure 5F*), confirming the results obtained with MB247Gal4 (*Figure 5C*). Therefore, αβ neurons are indispensable for habituation latency. Interestingly, silencing the α′β′ neurons with the highly selective split-Gal4 line MB463B (*Aso et al., 2014*) also facilitated habituation, but did not affect dishabituation (*Figure 5G*). Finally, silencing the γ neurons with the MB131B split-Gal4 (*Aso et al., 2014*) and the independent γ driver VT44966, did not affect habituation (*Figure 5H,I*).

Therefore, neurotransmission from αβ and α′β′, but not γ MB neurons and their activation ostensibly by the PNs upon odor exposure, is essential for stimulus value maintenance, which underlies normal habituation latency. Because only the αβ neurons are required for normal dishabituation with footshock, it appears that the dishabituating stimulus engages them differentially and distinguishes them functionally from their α′β′counterparts.

## Distinct neuronal subsets are required to establish olfactory habituation

Collectively, the results indicate that habituation latency is an active process requiring synaptic activity of distinct neuronal subsets engaged in odor information processing. However, of equal importance is timely habituation. To identify neurons mediating habituation we silenced the neuronal assemblies involved in the olfactory pathway and assessed avoidance after 4 min of OCT exposure, adequate for control flies to habituate (*Figure 1A*).

Blocking iLN or eLN antennal lobe interneurons was permissive to habituation after 4 min of OCT stimulation (*Figure 6A,B*). Perhaps, this is expected for the iLNs since their activity is required for response maintenance (*Figure 4A,B*), but not for the eLNs, which do not function in habituation latency (*Figure 4C*). We next investigated the role of the excitatory and inhibitory PNs marked by GH146Gal4 and MZ699Gal4, respectively. Surprisingly, given their role for habituation latency, silencing GH146 neurons blocked habituation after 4 min of OCT (*Figure 6C*). However, reducing GABAergic neurotransmission from the inhibitory GH146Gal4-marked neurons including the APL had no effect (*Figure 6—figure supplement 1*), indicating that the cholinergic neurons necessary for response maintenance also participate in establishing habituation. Significantly, blocking MZ699 neurons also eliminated habituation (*Figure 6D*), indicating that these neurons function specifically in the habituation and not during the preceding latency phase (*Figure 4F*). Therefore, both excitatory iACT and inhibitory mACT PN neurons are necessary to establish odor habituation. The performance of all remaining controls is presented in *Supplementary file 3*.

To verify the role of PNs in habituation and determine whether their activation is sufficient to mediate the process, we depolarized them artificially by activation of the heat-activated TRPA1 channel (*Pulver et al., 2009*) for 1 or 4 min without odor stimulation and then tested the flies for OCT avoidance. Activation of either PN subset for 1 min did not alter OCT avoidance (*Figure 6E,F*), as it did not when the GH146-marked PNs were depolarized for 4 min (*Figure 6E*). However, activation of the inhibitory MZ699-marked mACT PNs for 4 min attenuated OCT avoidance without prior

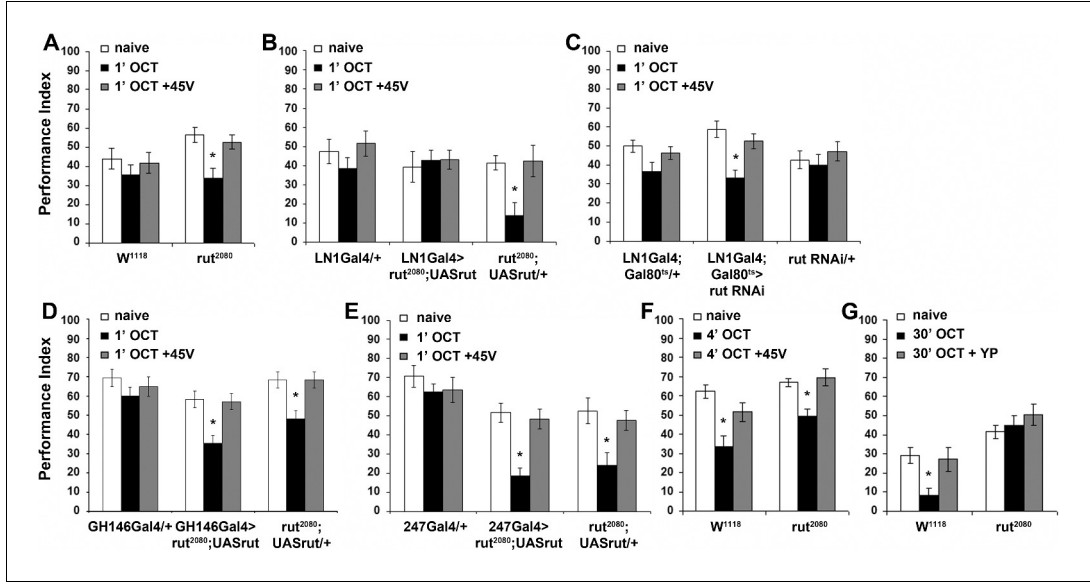

**Figure 7.** The Rutabaga Adenylyl Cyclase is required for olfactory habituation. Mean Performance Indices ± SEM are shown in all figures. Stars indicate significant differences from the naive response unless specified otherwise. (A) Rut[2080] mutants presented facilitated habituation to 1X OCT after only 1 min of exposure (black bar, p=0.0011) and normal dishabituation (grey bar, p=0.7873). Controls responded indistinguishably from naive animals to 1 min of 1X OCT (black bar, ANOVA p=0.5346). (n ≥ 14 for all groups) (B) Targeted Rut re-expression in rut[2080] mutants specifically in iLNs rescued their premature habituation (black bar, ANOVA p=0.8976), while rut[2080]; UASrut/+animals habituated prematurely (black bar, p=0.0079). (n ≥ 10 for all groups) (C) Rut abrogation via RNAi-mediating transgene expression in iLNs mimics the facilitated habituation of rut[2080] (black bar, p<0.0001), while both controls did not (LN1Gal4;Gal80[ts] ANOVA p=0.0606; rutabaga RNAi/+ANOVA p=0.5997). (n ≥ 18 for all groups) (D) Targeted Rut re-expression in rut[2080] mutants specifically in ePNs under GH146Gal4 did not rescue their premature habituation (black bar, p=0.0024), which is similar to that presented by rut[2080] mutants (black bar, p=0.0041). However, dishabituation was normal in both cases. (n ≥ 10 for all groups) (E) Targeted Rut re-expression in rut[2080] MBs under MB247Gal4 did not rescue the premature habituation (black bar, p<0.0001) in accord to the performance of rut[2080] mutants (black bar, p=0.0060). Dishabituation was normal in both cases. (n ≥ 10 for all groups) (F) rut[2080] habituated normally to 1X OCT after 4 min of exposure (black bar, p=0.0039) and dishabituated normally (grey bar, p=0.8037). (n ≥ 15 for all groups) (G) Rutabaga mutants did not habituate after 30 min 1X OCT exposure (black bar, ANOVA p=0.4287), in contrast to control flies that habituated normally (black bar, p=0.0067) and dishabituated with a yeast puff (grey bar, p=0.9279). (n ≥ 12 for all groups) Statistical details on *Supplementary file 1* and all data are presented in *Figure 7—source data 1*.
DOI: https://doi.org/10.7554/eLife.39569.028

The following source data and figure supplements are available for figure 7:

**Source data 1.** The Rutabaga Adenylyl Cyclase is required for olfactory habituation.
DOI: https://doi.org/10.7554/eLife.39569.031

**Figure supplement 1.** Habituation parameters after continuous exposure to OCT for 30 min.
DOI: https://doi.org/10.7554/eLife.39569.029

**Figure supplement 1—source data 1.** Habituation parameters after continuous exposure to OCT for30min.
DOI: https://doi.org/10.7554/eLife.39569.030

exposure to the odor (*Figure 6F*). These results suggest that prolonged activation of iPNs innervating the LH is necessary and sufficient for habituation, while the ePNs are also necessary, because silencing them blocks habituation (*Figure 6C*), but not sufficient to drive OCT habituation. It is likely that mACT PNs are functionally 'downstream' of those of the iACT in promoting habituation, possibly by inhibiting the LH-mediated innate odor avoidance.

Finally, we investigated the role of the MBs in habituation. Surprisingly, collective silencing of the MBs under MB247Gal4 resulted in normal habituation (*Figure 6G*). Therefore, the MBs are dispensable for establishing or expression of odor habituation, in contrast to their function during habituation latency.

## The rutabaga adenylyl cyclase is essential for normal olfactory habituation

The cAMP signaling pathway has been implicated in habituation of various modalities and circuits in Drosophila (*Engel and Wu, 2009*). Rutabaga, the major Adenylyl Cyclase in the adult *Drosophila* brain (*Han et al., 1992*), has been shown necessary for odor habituation after repetitive (*Cho et al., 2004*), or continuous odor exposure (*Das et al., 2011*). More specifically, Rut is required in LN1 neurons to mediate habituation after 30 min odor exposure (*Das et al., 2011*). Given the differences, we uncovered in the neuronal subsets implicated in habituation to 4 min and 30 min odor exposure, we investigated the role of Rut in the latency and habituation phases to OCT stimulation.

The $rut^{2080}$ (*Levin et al., 1992*) mutant presented premature habituation after only 1 min of OCT exposure (*Figure 7A*), suggesting a role for this protein in habituation latency. To elucidate whether Rut is indeed required for habituation latency within the LNs, PNs, or the MBs, where it is preferentially expressed (*Han et al., 1992*), we expressed a UAS-*rut* transgene in these neurons of $rut^{2080}$ mutants. Rut re-expression in LNs was sufficient to restore habituation latency after 1 min of OCT exposure (*Figure 7B*). This was independently validated by RNA interference (RNAi)-mediated Rut abrogation in adult LNs under LN1Gal4;Gal80$^{ts}$, which phenocopied the premature habituation of $rut^{2080}$ mutants (*Figure 7C*), confirming the role of Rut within the iLNs for habituation latency. In contrast, Rut re-expression in PNs (*Figure 7D*), or the MBs (*Figure 7E*), did not rescue the premature habituation phenotype of $rut^{2080}$ mutants, indicating that the protein is not required therein for response maintenance.

Habituation of $rut^{2080}$ mutants was normal after 4 min of OCT exposure (*Figure 7F*), indicating that Rut is not required for habituation. This is not unexpected, since Rut is required within the LNs, which are specifically required for habituation latency (*Figure 4A*), but are dispensable for habituation (*Figure 6A*). However, Rut has been reported essential for habituation to 30 min odor stimulation (*Das et al., 2011*). This difference is consistent with the notion that habituation to 4 min odor stimulation is distinct from habituation to 30 min of odor exposure, or it might be consequent of the different odor stimuli or experimental setups used in the two paradigms. To differentiate between these alternatives, we established habituation to 30-min OCT stimulation in our experimental setup. We demonstrate that as for 4-min habituation, continuous or pulsed OCT stimulation for 30-min results in response attenuation in control flies (*Figure 7—figure supplement 1A*). This long exposure habituated response recovered spontaneously after 6 or 30 min of rest (*Figure 7—figure supplement 1B,C*) in agreement with a prior report (*Das et al., 2011*). However, we were unable to dishabituate the long odor exposure habituation with footshock or vortexing (*Figure 7—figure supplement 1D,E*), as for habituation to 4 min of OCT. Interestingly, habituation after 30 min of OCT exposure was dishabituated with a short puff of yeast paste odor (*Figure 7—figure supplement 1F*, *Figure 7G*).

These results suggest that the response attenuation after prolonged odor stimulation is distinct from habituation to 4 min of exposure. Importantly, we verified that Rut is essential for habituation to 30-min OCT stimulation (*Figure 7G*), as has been shown by Das *et al* using different odors and experimental apparatus (*Das et al., 2011*). This indicates that Rut and by extension cAMP signaling, appear essential within inhibitory LNs, for habituation latency and prolonged odor habituation. This highlights the differential role of Rut in the dynamic engagement of circuits and molecular mechanisms therein, to ensure and regulate responses to continuous brief or prolonged inconsequential stimuli.

## Discussion

### A novel olfactory habituation paradigm

We describe a novel olfactory habituation paradigm to brief odor stimuli and operationally define two distinct phases in the response dynamics. The initial period of ~120 s we term habituation latency is characterized by maintenance of responsiveness to the odor. This is followed by manifestation of the habituated response, characterized behaviorally by attenuated osmotaxis. Focusing on the behavioral dynamics early in the process complements previous work olfactory habituation to continuous odor stimulation in *Drosophila* (*Das et al., 2011*; *Sadanandappa et al., 2013*). A number

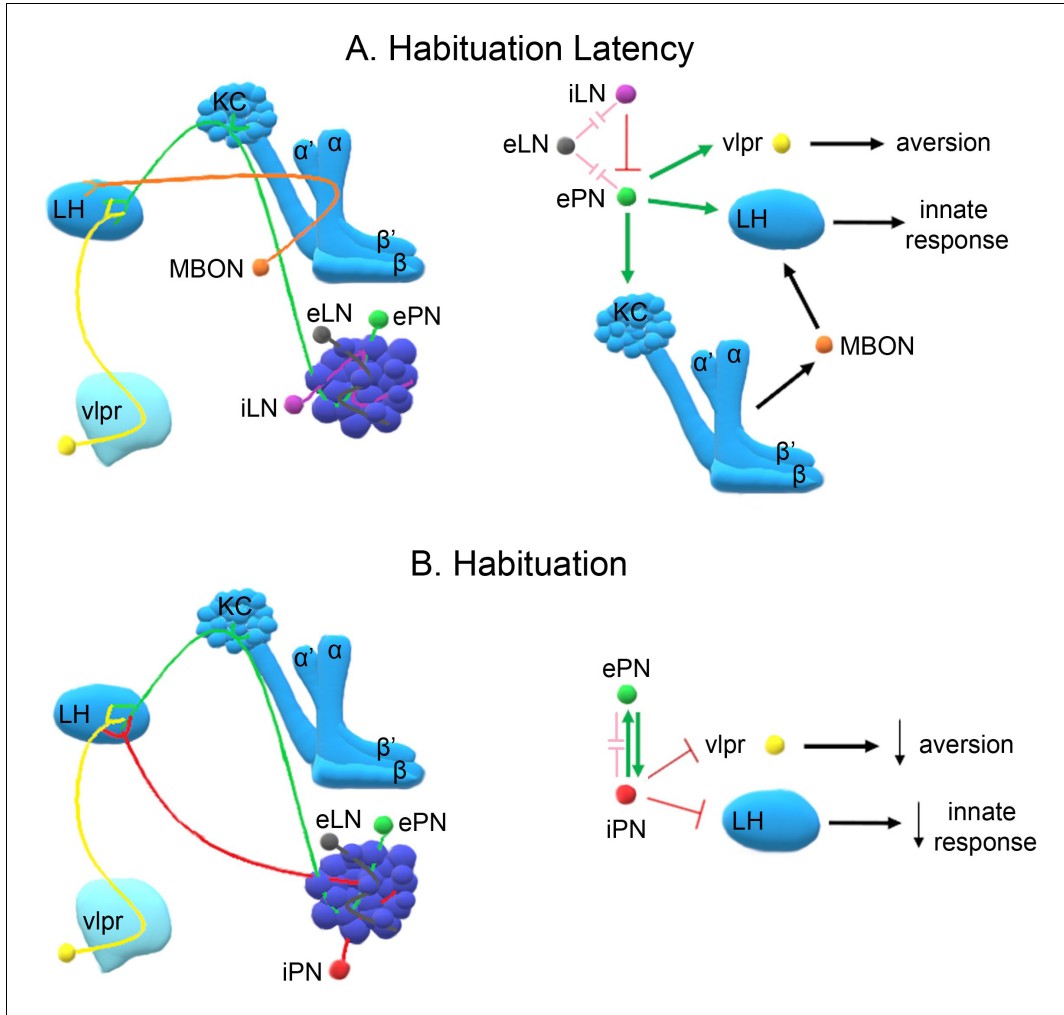

**Figure 8.** A model of the neuronal subsets underlying (**A**) Habituation Latency and (**B**) Habituation, after exposure to aversive stimuli. The antennal lobe, the mushroom bodies, the lateral horn (LH) and the ventrolateral protocerebrum (vlpr) are depicted with shades of blue, lighter blue showing higher order neurons. Distinct neuronal subsets are marked with different colors; iACT excitatory projection neurons (ePN-green), inhibitory local interneurons (iLN-purple), excitatory local interneurons (eLNs-grey), Mushroom Body Output Neurons (MBONs-orange), ventrolateral protocerebrum neurons (vlpr-yellow), mACT inhibitory projection neurons (iPNs-red). Green arrows indicate activation while red blunt arrows indicate inhibition. Pink blunt arrows indicate electrical synapses.
DOI: https://doi.org/10.7554/eLife.39569.032

of criteria differentiate these two paradigms from other types of habituation to olfactory stimuli as discussed below.

We show that *Drosophila* habituate equally well to continuous or pulsed olfactory stimuli (*Figure 1*, *Figure 1—figure supplement 1*, *Figure 7—figure supplement 1A*). This likely reflects the nature of olfactory stimuli, which typically are continuous rather than pulsed. On the other hand, habituation of the startle response to ethanol vapor (*Cho et al., 2004*) may specifically require short (30 s) pulses due to its sedative properties and this may also be reflected by the rather long 6 min ITIs compared to the 15 s to 2.5-min intervals used herein for OCT. Short odor pulses are also required for the odor-mediated jump and flight response habituation (*Asztalos et al., 2007*), suggesting that pulsing may be necessary to evoke the startle response per se.

An important property shared with all habituation paradigms in Drosophila and other systems is spontaneous recovery of the response (*Thompson and Spencer, 1966*; *Rankin et al., 2009*). This is another differentiating parameter among habituation paradigms in Drosophila. For the olfactory habituation paradigms, whereas 6 min suffice for spontaneous recovery after 4 and 30 min

continuous odor exposure (*Figure 2A,B* and *Figure 7—figure supplement 1B,C*), 15–30 (*Cho et al., 2004*) to surprisingly 60 min (*Asztalos et al., 2007*) are required for recovery in the olfactory startle paradigms. Habituation to mechanosensory stimuli typically also requires shorter spontaneous recovery times, with habituation of the giant fiber-mediated jump-and-flight response requiring a mere 2 min (*Engel and Wu, 1996*) and electric footshock habituation 6 min (*Acevedo et al., 2007a*). Interestingly, other non-mechanosensory habituation paradigms require long spontaneous recovery periods with 30 min for habituation of the proboscis extension reflex (PER), (*Paranjpe et al., 2012*) and surprisingly, 2 hr for habituation of odor-induced leg response (*Chandra and Singh, 2005*). We posit that these differences reflect the engagement of distinct neuronal circuits mediating habituation to these diverse stimuli and the properties and connections of the neuronal types that comprise them.

## Olfactory habituation phases are mediated by distinct neuronal subsets

Overall, our data suggest that latency and habituation to brief odor exposure involve modulation of LH output, a neuropil innately encoding response valence to odor stimuli (*Fişek and Wilson, 2014*; *Schultzhaus et al., 2017*). We propose that habituation latency involves processes that are not permissive to, or actively prevent stimulus devaluation. Latency duration depends on stimulus strength (*Figure 2D*), as suggested (*Thompson and Spencer, 1966*; *Rankin et al., 2009*) and is consistent with the notion that it is adaptive not to devalue strong, hence potentially important stimuli, expediently. In fact, we posit that habituation latency serves to facilitate associations with concurrent stimuli, a requirement for associative learning. Shortened latency leading to premature habituation is predicted to compromise associative learning.

Importantly, maintaining responsiveness early upon odorant exposure requires activity of GABAergic inhibitory neurons (*Figure 4A,B*), which are essential for lateral inhibition of antennal lobe glomeruli (*Figure 8A*). LN activation appears to prevent saturation by strong continuous odors and hence reduce PN activity (*MacLeod and Laurent, 1996*; *Olsen et al., 2010*). Therefore, shortening habituation latency by blocking GABAergic neurotransmission in the antennal lobe may effectively reduce stimulus intensity, expediting habituation as suggested by the dilute odor experiments (*Figure 2D*). This interpretation is further supported by the decreased habituation latency upon silencing the iACT PNs (*Figure 4D*) conveying olfactory signals to the MBs and the LH (*Tanaka et al., 2004*; *Stocker et al., 1997*), but not by the mACT neurons (*Figure 4F*) innervating only the LH (*Liang et al., 2013*). Since iACT PNs are mainly excitatory, it appears that response maintenance requires excitatory signaling to the LH and the MBs (*Figure 8A*).

All MB neuronal types except the γ, are essential for habituation latency (*Figure 5*). This suggests that at least part of the excitatory signal conveyed by the iACT PNs impinges upon the αβ and α′ β′ MB neurons (*Tanaka et al., 2004*; *Stocker et al., 1997*), which is consistent with their role in associative learning (*Busto et al., 2010*) and the proposal that habituation latency facilitates it. Neurotransmission from the MBs to LH neurons mediating aversive responses likely engages MB output neurons (MBONs), to maintain the valence and intensity of the odor and sustain aversion (*Figure 8A*). Distinct MBONs are known to drive both attraction and aversion to odors (*Aso et al., 2014*) and their potentially differential involvement in habituation is currently under investigation.

Dishabituation results in stimulus value recovery and apparently resets habituation latency (*Figure 3C*). Clearly it requires neurotransmission via the GH146-marked neurons and MBs because silencing these neurons disables dishabituation (*Figure 4D*, *Figure 5C,D,F*, *Figure 6G*), consistently with their role in response maintenance. These results lead us to hypothesize that dishabituating stimuli might converge on the MBs and/or iACT, possibly stimulating excitatory neurotransmission to the LH, to reinstate stimulus aversion. This hypothesis is currently under investigation as well.

In contrast, habituation requires prolonged or repeated exposure to the odorant and functional iACT and mACT PNs (*Figure 8B*) converging on the LH (*Tanaka et al., 2004*; *Liang et al., 2013*). Interestingly, the mainly GABAergic mACT PNs (*Okada et al., 2009*) receive input both from the olfactory sensory neurons and the excitatory iACT PNs (*Wang et al., 2014*). Their depolarization also activates the excitatory iACT neurons via direct chemical synapses (*Shimizu and Stopfer, 2017*). This apparent feedback loop may be required for mACT activation after prolonged exposure to aversive odors, since these neurons were reported to respond mainly to attractive stimuli (*Strutz et al., 2014*). We propose that prolonged aversive odor exposure enhances iACT activation, which in turn leads to habituation (*Figure 6F*, *Figure 8B*), while shorter exposure does not activate the iACT neurons, reflected by their dispensability for habituation latency (*Figure 4F*). Importantly,

the mACT innervates the LH downstream of the iACT PNs, providing feedforward inhibition (*Wang et al., 2014*). These characteristics likely underlie the necessary and sufficient role of mACT PNs in habituation upon 4-min odor stimulation. Collectively, our results are consistent with the proposal that mACT activation inhibits the innate LH-mediated avoidance response to the aversive odorant, establishing habituation (*Figure 8B*). However, full mACT activation appears to also require iACT neurotransmission, which if abrogated eliminates habituation (*Figure 6C*) but is insufficient to establish it on its own (*Figure 6E*).

Because the MZ699 Gal4 driver also marks ventrolateral protocerebrum (vlpr) neurons it is possible that they also play a role in habituation. In fact, vlpr neurons function in aversive odor responses, are activated by excitatory iACT PNs, inhibited by the inhibitory PNs, and are afferent to the LH (*Liang et al., 2013*). Thus, they could act in parallel or synergistically to mACT PNs to establish the habituated response. As we are not aware of a specific vlpr driver, it is impossible at the moment to address this possibility directly. Briefly then, our collective results strongly suggest a novel role for the inhibitory PNs innervating the LH, and possibly vlpr neurons, in inhibition of the innate response and habituation (*Figure 8B*). The kinetics of inhibitory projection neuron activation and their output on downstream neurons could serve as a measure of the duration of odor exposure. Upon prolonged exposure, these neurons mediate inhibition of odor avoidance, thus devaluing the stimulus.

Analysis of the neuronal subsets underlying habituation has focused on aversive odors. However, considering the neuronal clusters involved in the process, it would be relatively safe to assume that our results extend to attractive odor habituation as well. It is possible that the neuronal circuitry comprised of PNs, the LH and MBs may be mediating habituation independently of odor valence. However, specific neuronal clusters may differ in odor valance-dependent activation or inhibition of other circuit components with opposing effects on the behavioral readout. For example, inhibitory PNs (iPNs) mediate attraction by releasing GABA in the LH to inhibit avoidance (*Strutz et al., 2014*). If inhibited themselves, the resultant attenuated attraction will likely drive a behavioral output of habituation to an attractive odor.

In accord with this notion, attractive and aversive odors are represented in different AL glomerular clusters (*Seki et al., 2017*) and this valence-dependent organization is preserved into higher brain centers. In fact, the posterior-dorsal LH responds to attractive and its ventral complement to aversive odors (*Seki et al., 2017*), while third order neurons convey information from ventral LH to the vlpr (*Liang et al., 2013*; *Strutz et al., 2014*) and from the dorsal LH to the superior medial protocerebrum (*Fişek and Wilson, 2014*). This organization potentially reflects differential recruitment of these neuronal clusters in habituation to aversive and attractive odors. The circuits involved in habituation to attractive odors and their specific contribution to the process will be the focus of future work.

## Habituation after prolonged odor stimulation

Although behaviorally there is significant osmotactic attenuation after both 4 and 30 min aversive odor exposure, our experiments suggest that these represent distinct types of olfactory habituation. Habituation after 4 min of odor exposure does not require the MBs, but rather the projection neurons innervating the LH (*Figure 6C,D*, *Figure 6F*). Habituation after 30 min of exposure is also independent of MB function (*Das et al., 2011*), but appears to be entirely mediated by iLNs and reside within the AL (*Das et al., 2011*). This clear difference suggests that the specific potentiation of inhibitory synapses shown to underlie habituation after 30 min of exposure is not necessary for habituation to the brief 4-min exposure. Additionally, while Rut is required within the iLNs during the latency period upon brief odor exposure (*Figure 4A,B*), it is surprisingly required within the same neurons for habituation to long odor exposure (*Das et al., 2011*). Therefore, Rut-driven activity within the iLNs yields opposing time-dependent behavioral outputs in accord with the abovementioned notion that the same circuit components may drive opposing outputs.

Furthermore, the fact that mechanosensory stimuli are not effective dishabituators after 30 min of odor exposure as they are after 4 min (*Figure 7—figure supplement 1D,E*), augments the conclusion these are different types of olfactory habituation and suggests that distinct dishabituators likely recruit different neuronal subsets to modulate the habituated response. Such neuronal circuits and the effect of different dishabituators in response recovery are currently under investigation.

Altogether, the results indicate different mechanisms for 4 min and 30 min habituation to aversive odors with the former mediated by the interaction between iPNs, ePNs and their targets in the LH,

while the latter is based on the inhibition of ePNs by iLNs at the AL level. However, it is possible that the potentiated PN inhibition would decrease their output to the LH to drive reduced avoidance. This argues that the LH could be involved in the behavioral output indicating habituation after 30 min of OCT exposure as well. An AL-mediated reduction in the perceived intensity or valence of a chronically present odor probably serves an adaptive evolutionary role distinct from short exposure to the same stimulus. In fact, filtering away the chronic odor at the antenna, the first olfactory synaptic station, might facilitate evaluation of additional odors at higher order neurons such as the MBs or the LH.

Significantly, this interpretation is congruent with timescale habituation in mice, where short-timescale odor habituation is mGluR-dependent and mediated by the anterior piriform cortex while long-timescale habituation requires NMDAR and is mediated by the olfactory bulb (*Chaudhury et al., 2010*). In addition, studies in mice, rats and primates have shown that habituation of the higher order neurons is faster and more prominent than in olfactory bulb neurons (*Zhao et al., 2015*; *Zhao et al., 2016*). Therefore, temporal and spatial principles for olfactory habituation appear broadly conserved between insects and mammals, despite their evolutionary distance.

## Materials and methods

### *Drosophila* strains

*Drosophila* were cultured in standard wheat-flour-sugar food supplemented with soy flour and $CaCl_2$ at 22–25C, unless specified otherwise. Animals expressing Gal80$^{ts}$ (TARGET system) were raised at 18C until hatching and then placed at 30C for 3 days prior to testing. Animals expressing the tetanus toxin light chain transgene (UAS-TTX) (*Sweeney et al., 1995*) were raised at 18C until hatching, then placed at 21–22C for 2 days prior to testing. $TT_{LC}$ cleaves synaptobrevin, a protein required for synaptic vesicle docking, thus silencing presynaptic neurons constitutively (*Humeau et al., 2000*).

The control strain Berlin and the *mushroom body miniature* (*mbm$^1$*) mutants have been described previously (*Heisenberg et al., 1985*). Control flies carrying the $w^{1118}$ mutation had been backcrossed to the Canton-S for at least 10 generations ($w^{1118}$ strain). As a second control, $w^*$, an independent mutation in the *white* gene was used. All other strains had been backcrossed to the Cantonised-$w^{1118}$ for four to six generations prior to use in behavioral experiments. The *rutabaga* mutant $rut^{2080}$ was described previously (*Levin et al., 1992*). Transgenes used to block neurotransmission were the UAS-TTX (*Keller et al., 2002*), encoding the tetanus toxin light chain and the UAS-shi$^{ts}$, which bears a temperature-sensitive mutation in dynamin, encoded by the gene *shibire* (*Kitamoto, 2001*). To achieve neuronal hyperpolarization, the transgene UAS-TRPA1 was used for overexpression of the TRPA1 channel (*Rosenzweig et al., 2005*).

GH146, MB247, c772 and c739 Gal4 drivers have been described before (*Acevedo et al., 2007b*; *Pavlopoulos et al., 2008*) and similarly for OK72 (*Acebes and Ferrús, 2001*). MB463B and MB131B targeting the α′β′ and γ lobe, respectively, were kindly provided by Y. Aso (Janelia Research Campus, Howard Hughes Medical Institute, Ashburn, VA) (*Aso et al., 2014*). The VT44966 γ lobe driver was obtained from the Vienna Drosophila Resource Center (VDRC, #203571), as was the UAS-Gad RNAi (VDRC, #32344). The APLGal4 driver was described previously (*Wu et al., 2013*). The UAS-rutabaga was obtained from Bloomington (#9405) and has been described before (*Zars, 2000*). MBGal80 was obtained from Ron Davis (Scripps Florida), while LN1Gal4, LN1Gal4;Gal80$^{ts}$ (*Sudhakaran et al., 2012*) and UAS-rutabaga RNAi (VDRC#5569) (*Das et al., 2011*) were provided by M. Ramaswami (Trinity College, University of Dublin, Dublin, Ireland). GH298Gal4 (*Stocker et al., 1997*) was obtained from A. Ferrus (Instituto Cajal, C.S.I.C., Madrid, Spain) and krasavietzGal4 (*Shang et al., 2007*) was provided by A. Fiala (Georg-August-Universität Göttingen, Göttingen, Germany). MZ699Gal4 (*Tanaka et al., 2012*) was kindly provided by Liqun Luo (Department of Biological Sciences Stanford University, Stanford, CA).

### Hydroxyurea treatment

Hydroxyurea ablation of the MBs was achieved using 75 mg/ml hydroxyurea (HU) as described previously and each batch of HU-treated adults was monitored histologically for the extent of mushroom body ablation before using flies from the particular brood for testing (*Acevedo et al., 2007b*).

## Behavioral analyses

To obtain flies for behavioral analyses, Gal4 driver homozygotes were crossed *en masse* to UAS-shi[ts], UAS-TTX and UAS-TRPA1. Similarly, UAS-shi[ts], UAS-TTX, UAS-TRPA1 and Gal4 driver homozygotes were crossed *en masse* to $w^{1118}$, to obtain heterozygous controls. For the rescue experiments virgins $rut^{2080}$;UAS-rutabaga were crossed to Gal4 driver homozygotes or $w^{1118}$. Since the *rutabaga* gene is on the X chromosome, behavior was tested only in male mutant and control flies. For the rutabaga RNAi experiment LN1Gal4;Gal80[ts] homozygotes were crossed to UAS-rutabaga RNAi homozygotes or $w^{1118}$. UAS-rutabaga RNAi homozygotes were also crossed to $w^{1118}$ to obtain the heterozygous control flies. For the Gad RNAi experiments, GH146Gal4 homozygotes were crossed to UAS-Gad RNAi and $w^{1118}$, while $w^{1118}$ flies were crossed to UAS-Gad RNAi to obtain heterozygous controls. Flies for all experiments were raised at 25C, except for the neuronal activation experiments with UAS-TRPA1 and the rutabaga RNAi experiment, where flies were raised at 18C. All flies used in behavioral experiments were tested 3–5 days after emergence. Behavioral experiments were performed under red light at 23–24C and 60–70% humidity.

### Osmotaxis

Odor avoidance and attraction were quantified by exposing approximately 50 flies at the choice point of a standard T-maze (*Acevedo et al., 2007b*) to an air-stream (500 ml/min) carrying the odor in one arm and fresh air in the other. The odorants utilized for these experiments were 1000 µl of 3-octanol (OCT) (Acros), 100 µl of benzaldehyde (BNZ) (Sigma), 10 µl of a 0.1% dilution in water of ethyl acetate (ETA) (Sigma) and 10 µl of a 0.5% dilution of 2,3-butanedione (Sigma). Flies were given 90 s to choose between aversive odors and air. In contrast, control experiments (not shown) determined that 180 s for the choice between attractive odors and air gave the most consistent and reliable indices. At the end of the choice period, flies in each arm were trapped and counted. The performance index (PI) was calculated as the percentage of the fraction of flies that avoid the odor and congregate in the unscented (air) arm minus the fraction of flies that prefer the odor-bearing arm.

### Olfactory habituation

Olfactory habituation experiments were performed under the conditions described above. For the 'training phase', approximately 50 flies were exposed in the upper arm of a standard T-maze to either attractive (ETA, BUT) or aversive odors (OCT, BNZ) for the indicated times. After a 30 s rest period, the flies were lowered to the center of the maze for testing their osmotactic response by a choice of air vs. either the previously experienced, or a novel odor. At the end of the choice period (90 for aversive and 180 s for attractive odors), the flies in each arm were trapped and counted and the performance index was calculated as described above. UAS-shi[ts] harboring strains were placed in a 32C incubator for 30 min prior to the start of the 'training phase' to inactivate the transgenic temperature-sensitive Shibire protein, which recovers its full activity within 15 min after removal from 32C (*Kitamoto, 2001*; *McGuire et al., 2001*). Flies overexpressing the wild-type rutabaga gene in the $rut^{2080}$ mutant background, and flies overexpressing the UAS-Gad RNAi, were kept at 30C overnight, to maximize trangene expression. To examine spontaneous recovery, flies were given rest periods of variable lengths as indicated, 6 min being the experimentally derived standard recovery time, within the upper arm of the maze after 4 min odor pre-exposure. For the spontaneous recovery after 30 min of pre-exposure, flies were returned to food vials for the 6 or 30 min rest period. Subsequently, they were tested against the odor they were pre-exposed to, versus air.

For the pulsed odor stimulation, we divided the 1 min of continuous odor exposure into two 30 s odor pulses with an 8 s interstimulus interval (ITI), the 4-min odor exposure into four 1 min odor pulses with the proportional 15 s ITI and finally the 30-min exposure was divided into three 10 min pulses with 2.5 min ITI. ITI length was kept at a quarter of the odor exposure duration, since adaptation has been correlated with stimulus duration (*de Bruyne et al., 1999*).

To determine the conditions for dishabituation with electric shock, control experiments were performed first to determine the stimulus strength and number of shocks required. Dishabituation was attempted at different shock stimulus strengths with the following results: OCT PI for naive: 59.2 ± 2.3. Habituated OCT PI: 20.6 ± 1.7. Dishabituation with 30 V, OCT PI: 54.8 ± 2.1; with 45 V, OCT PI: 58.7 ± 2.6; with 90V OCT PI: 58.3 ± 2.2. Moreover, the number of shocks did not have a

significant effect on dishabituation (1 × 90V shock OCT PI: 58.7 ± 2.3; 2 × 90V shock OCT PI: 59.4 ± 2.8). Since the 90V and 45V dishabituating shocks had equal effects, the weaker of the two was selected.

Vortexing was used as another mechanical stimulus to produce dishabituation. Flies were subjected to 3 s of vortexing at maximal speed immediately after odor exposure. Finally, for the 30 min odor pre-exposure habituation experiments flies were exposed to air drawn at 500 ml/min over a 30% (w/v) aqueous solution of Brewers yeast (Acros Organics) for 3 s after the odor pre-exposure to dishabituate. The performance index (PI) for habituation and dishabituation was calculated as described above.

For neuronal activation experiments, expression of UAS-TRPA1 was driven to the neurons of interest. Flies used in activation experiments were raised at 18C and expression of the transgene was induced at 31C (using a heat block), for 1 min or 4 min prior to testing.

## Statistical analysis

Untransformed (raw) data were analyzed parametrically with JMP3.1 statistical software package (SAS Institute Inc., Cary, NC). If significant, initial ANOVA tests were followed by Dunnett's and Least Square Means Contrast analyses and the experimentwise error rate was adjusted as suggested by Sokal and Rohlf (Appendix B) (*James Rohlf and Sokal, 2012*). Detailed statistics are found on *Supplementary file 1*.

## Acknowledgements

We thank R Davis (The Scripps Research Institute), M Ramaswami (University of Dublin), Y Aso (Howard Hughes Medical Institute), Gregg Roman (University of Mississippi), L Luo (Standford University), AK Kanellopoulos (University of Lausanne) and the Bloomington Stock Center for fly strains. This work has been co-financed by the European Union (European Social Fund – ESF) and Greek National funds through the Operational Program 'Education and Lifelong Learning' of the National Strategic Reference Framework 2007–2013, Research Funding Program: THALES- Investing in knowledge society through the European Social Fund, MIS: 376898. We also acknowledge support by the project 'Strategic Development of the Biomedical Research Institute 'Alexander Fleming'' (MIS 5002562) which is implemented under the 'Action for the Strategic Development on the Research and Technological Sector', funded by the Operational Programme 'Competitiveness, Entrepreneurship and Innovation' (NSRF 2014–2020) and co-financed by Greece and the European Union (European Regional Development Fund). Parts of this work were also supported by Fondation Sante.

## Additional information

### Funding

| Funder | Grant reference number | Author |
| --- | --- | --- |
| Hellenic Secretariat of Research and Technology | MIS 376898 | Ourania Semelidou Efthimios MC Skoulakis |
| Biomedical Sciences Research Centre "Alexander Fleming" | MIS 5002562 | Ourania Semelidou Efthimios MC Skoulakis |
| Fondation Santé | | Ourania Semelidou Efthimios MC Skoulakis |

The funders had no role in study design, data collection and interpretation, or the decision to submit the work for publication.

### Author contributions

Ourania Semelidou, Conceptualization, Formal analysis, Validation, Investigation, Methodology, Writing—original draft; Summer F Acevedo, Conceptualization, Investigation, Methodology; Efthimios MC Skoulakis, Conceptualization, Supervision, Funding acquisition, Methodology, Project administration, Writing—review and editing

## Author ORCIDs

Ourania Semelidou (iD) http://orcid.org/0000-0001-8774-6013
Efthimios MC Skoulakis (iD) http://orcid.org/0000-0001-5113-6192

## Decision letter and Author response

Decision letter https://doi.org/10.7554/eLife.39569.040
Author response https://doi.org/10.7554/eLife.39569.041

## Additional files

### Supplementary files

• Source data1. performance of driver heterozygotes upon 1 min odor exposure.
DOI: https://doi.org/10.7554/eLife.39569.033

• Source data 2. Performance of driver heterozygotes upon 4 min odor exposure.
DOI: https://doi.org/10.7554/eLife.39569.034

• Supplementary file 1. Collective statistical analyses and results for all data in the paper.
DOI: https://doi.org/10.7554/eLife.39569.035

• Supplementary file 2. Responses of control animals, heterozygous for the Gal4 drivers when naive, after 1 min pre-exposure and after 1 min pre-exposure followed by one 45V electric footshock application. Data are represented as mean ± SEM and all data are presented in *Source data 1.* subsequent Dunnett's test: p=0.0016 for 1 min OCT and p=0.9918 for 1 min OCT +45V two subsequent Dunnett's test: p=0.0003 for 1 min OCT and p=0.1084 for 1 min OCT +45V
DOI: https://doi.org/10.7554/eLife.39569.036

• Supplementary file 3. Responses of control animals, heterozygous for the Gal4 drivers when naïve, after 4 min pre-exposure and after 4 min pre-exposure followed by one 45V electric footshock application. Data are represented as mean ± SEM and all data are presented in *Source data 2.* subsequent Dunnett's test: p<0.0001 for 4-min OCT and p=0.9061 for 4-min OCT +45V two subsequent Dunnett's test: p<0.0001 for 4 min OCT and p=0.1594 for 4-min OCT +45V three subsequent Dunnett's test: p<0.0001 for 4-min OCT and p=0.6941 for 4-min OCT +45V four subsequent Dunnett's test: p=0.0007 for 4-min OCT and p=0.7259 for 4-min OCT +45V five subsequent Dunnett's test: p<0.0001 for 4-min OCT and p=0.1046 for 4-min OCT +45V
DOI: https://doi.org/10.7554/eLife.39569.037

• Transparent reporting form
DOI: https://doi.org/10.7554/eLife.39569.038

### Data availability

All data generated or analysed during this study are included in the manuscript and supporting files. Source data files accompanying each figure and Supplementary files have been uploaded.

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
