## [Decision Letter]

[**Editorial note:** This article has been through an editorial process in which the authors decide how to respond to the issues raised during peer review. The Reviewing Editor's assessment is that all the issues have been addressed.]

Thank you for submitting your article "Dynamic engagement of distinct neuronal circuits regulating Olfactory Habituation in *Drosophila*" for consideration by *eLife*. Your article has been reviewed by three peer reviewers, and the evaluation has been overseen by Mani Ramaswami as Reviewing Editor and Eve Marder as the Senior Editor. The following individuals involved in review of your submission have agreed to reveal their identity: Alberto Ferrus (Reviewer #1); David L Glanzman (Reviewer #3). Reviewer #2 remains anonymous.

The Reviewing Editor has summarised the concerns that require revision and/or responses, and we have included the separate reviews below for your consideration. If you have any questions, please do not hesitate to contact us.

The work is interesting as it provides evidence to support the idea that shorter-term forms of habituation (as well as an unexpected resistance to early habituation) operate through different molecular and neural circuit mechanisms. Thus, the authors describe original and new phenomena that should make one think differently about different phases of habituation, their regulation, function and interaction with mechanisms of associative learning and memory.

Overall concerns:

The paper will be improved overall by including a more complete discussion of how the different phases and mechanisms of habituation operate in concert (even if this involves some speculation) integrating these findings into the existing literature on habituation, not only in *Drosophila*, but also, where relevant, in other organisms. In addition the paper could be overall better communicated by carefully choosing terms most appropriate to the message of the paper, and also are consistent with previous usage. The Abstract, Introduction and Discussion would benefit from a deep re-write, to accommodate revisions as well as to clarify that the new findings pertain to (very) short term habituation and not to longer-term forms of this process.

The paper will also be improved by addressing each of the comments detailed in the individual reviews below. For example, by considering the involvement of sensory adaptation in their 1-, 4-, and 30-min exposure protocols; and the potential effects of repeated but briefer exposures (e.g. 4 x 1-min vs 1 x 4-min).

The reviewer discussion also noted the need to explain the observation that silencing GH146 neurons does not affect the behavior of naïve flies to OCT (Figure 4D), which is potentially confusing given that OCT is sensed at least in large part by GH146 expressing neurons.

Separate reviews (please respond to each point):

*Reviewer #1:*

The manuscript reports a novel method to evaluate habituation to odorant stimuli in *Drosophila*. In essence, the new method uses a 4 min odorant pre-exposure in contrast to the 30 min protocol used in other labs. Genetic manipulations to silence or excite specific subsets of neurons in the olfactory pathway are used here to identify groups of cells involved in the different phases of the 4 min habituation. The results clearly point towards a diversity of mechanisms which render habituation as a rather heterogeneous learning process. This is an idea shared by many colleagues in the field which makes the manuscript welcome.

There are, however, some issues that need to be addressed, in particular when drawing functional interpretations and circuits:

1) In addition to chemical synapses, neurons of the olfactory pathway are connected by electrical coupling. Actually, this coupling is stronger than the chemical communication between eLN and PN according to studies previously published (albeit not quoted) (Yaksi et al., 2010; Huang et al., 2010). Electrical coupling is also functional, although weaker than the chemical communication, between eLN and iLN, and among PN neurons. In the experiments in which tetanous toxin or shibire are used to silence specific groups of neurons, the electrical synapses remain unaffected, hence communication between the referred groups of neurons must be still operational. This condition surely will affect the interpretation of the results summarized in Figure 8.

2) The conclusion that the 30 min and 4 min habituations are mechanistically (Rut dependence) and spatially (LH versus AL) different processes is attractive, but, given the large difference in the times involved, it appears counterintuitive. How could a 30 min process be kept constrained within the AL?

3) The use of concentrated and diluted odorant solutions is considered as "strong" and "weak" stimuli here and throughout the literature. Being strict, however, we cannot use these terms in the absence of electrophysiological evidence showing higher frequency firing in the first case with respect to the second. If the key neurons involved would be of the low responder type (firing sparse action potentials only), a diluted odorant stimulus would elicit the same response as a concentrated one. At least the MB neurons belong to the sparse responder type.

4) The use of Gal4 drivers is quite extended in *Drosophila* neurobiology and the specificity of their domains of expression is strongly emphasized. Detailed scrutiny of these expression domains, however, shows far more variability than usually admitted. For instance, based on the GTRACE technique most Gal4 drivers exhibit variable expression during development and among individuals (Evans et al., 2009). One driver used here, GH298-Gal4, has been reported to exhibit transient expression in neurons of the adult brain (Casas-Tinto et al., 2017).

All these comments do not diminish the value and general interest of the manuscript but represent issues that should be considered in the interpretation of the data.

Finally, a few minor points on typos:

Materials and methods section, third from last paragraph: 3econds >> 3 seconds

Subsection “Hydroxyurea treatment”: extend >> extent

Discussion paragraph nine: are consistent the >> consistent with

*Reviewer #2:*

This paper reports a behavioral protocol to study decrement in responses to aversive or attractive odors following prior exposure to the same or a different odor. The results are novel and potentially interesting to a broad readership. However, before the paper is accepted for publication, several fundamentally important issues must be resolved or carefully qualified to avoid confusion in the literature.

1). Sensory adaptation and olfactory habituation. The olfactory behaviors described here do conform with some of the criteria of Habituation, including response decrement following prior exposure, spontaneous recovery, and dishabituation (cf. Rankings et al., 2009, Neurobiol. Learn Mem, not cited). However, a commonly adopted protocol of repetitive stimulus delivery is missing here.

Instead of the repetitive nature of stimulation in other paradigms, relatively prolonged, continuous, exposures to the odors OCT, ETA or BNZ were adopted. While the fly would take only fractions of a second or seconds to respond, it endures continuous, passive exposure to the odors for minutes in this protocol. It makes one wonder how much sensory adaptation takes place during the prolonged 1-, 4-, or even 30-min exposure? Earlier research from Siddiqi's and others' times needs to be carefully reviewed. How do the OSN spikes or even EAG recordings change over such time scales, i.e. how much receptor adaptation actually occurred during minutes of exposure?

2) The authors have not made a convincing case that an optimal and relevant protocol has been worked out through a comprehensive, systematic experimental trials. The current approach has a relatively poor temporal resolution. Empirically, varying or modifying some simple, basic experimental conditions will produce new, useful information. For example, how do the outcomes differ for the prior exposure schemes of 2X30sec vs 1X1min? 4X1min vs 1X4min? or 3X10min vs 1X30min (Figures 7G and Figure 7—figure supplement 1D-F)? In other words, is the cumulative exposure time or repetition exposures more effective?

3) In Figures 7G and Figure 7—figure supplement 1D-F, rutabaga mutants did not habituate after 30 minutes OCT exposure whereas WT displayed clear habituation but its "dishabituation" required "yeast puff". The results are intriguing but should be presented in the context of distinguishing different forms of sensitization. Why does electric shock or mechanical vortexing no longer work? What would be the time course of spontaneous recovery for this form of long-term "olfactory habituation"?

4) The authors propose that there are two phases of their habituation process, which could be dissected in genetic manipulations and be associated with different identified neural circuits. The proposed concepts of "refractory period maintenance" and "habituation onset" appear to be mostly based on the differences between 1min vs 4 min exposure and the rut/AC mutantational effects. Is this truly the only interpretation? Could it be two separate kinds of sensory processing with kinetics over different time scales/ temporal domains?

5) The authors may want to review and compare their results with the previous *Drosophila* habituation paradigms for aspects of repeated stimuli, spontaneous recovery, and dishabituation. A more global picture may emerge to enhance the presentation and strengthen their contributions to the field. A partial list:

i) The giant fiber-mediated jump & flight escape/startle reflex (repetitive light flash / air puff dishabituation)ii) Proboscis extension reflex (also in honey bees).iii) Olfactory startle jump reflex (repetitive odor puffs / vortex dishabituation)iv) Landing response (repetitive visual looming / vibration dishabituation)

In particular, habituation process of the escape reflex following startling stimuli and other processes also involves dunce and rutabaga. These lines of observations may converge upon a short time scale process that is related to the relevant sensory information processing.

*Reviewer #3:*

This is an interesting paper on the neural mechanisms underlying a novel form of short-term olfactory habituation in *Drosophila*. The experiments have been carefully performed and the data appear reliable. However, in my opinion the paper is flawed by the terminology used to describe the basic behavioral phenomena; the terms are poorly chosen and make the findings unnecessarily difficult for the reader to understand. Below are my recommendations for improving the paper.

Major comments:

1) The paper identifies and analyzes two basic aspects of the olfactory habituation: the minimum time of exposure to an odorant of a given concentration sufficient to trigger habituation, which is termed the "refractory period"; and the decrease in the flies' responsiveness to an odorant, which is termed "habituation onset".

First, refractory period is a poor choice because it connotes two discrete exposures to a stimulus, for example the refractory period between the ability of two depolarizing current pulses to evoke an action potential in a neuron. By contrast, the odorant exposure used to determine the minimum exposure time required to elicit habituation is continuous. This phenomenon should be given a more accurate name, such as "critical time of stimulus exposure" or "minimum (stimulus) exposure time sufficient for habituation induction".

Second, what the authors mean by "habituation onset" is simply whether or not the phenomenon of habituation is present; they are not referring specifically to the onset of habituation, but, rather, to habituation itself. The reference to "onset" is confusing. The authors should simply use the word "habituation" or, where appropriate, "induction of habituation".

2) It is claimed that the form of habituation reported in the paper lacks the stimulus generalization that characterizes many (most?) forms of habituation, as originally noted by Thompson and Spencer. But the basis for the claim here that the habituation does not generalize is a comparison between two odors of the same (negative) valence and two odors of opposite valence. These are hardly exhaustive tests for the presence of stimulus generalization; the authors should modulate their claim accordingly.

3) Initially, habituation to attractive odorants, ethyl acetate (ETA) and 2,3-butanedione (BUT) is examined; however, after the experiments presented in Figure 1 are summarized, habituation to attractive odorants is not discussed further. Therefore, the extent the cellular models presented in Figure 8 pertain to attractive, as well as repellant, odorants is unclear. This point should be discussed.

Minor Comments:

1) The classic paper by Thompson and Spencer, 1966 is cited. However, the more recent paper by Rankin et al., 2009, which presents an updated discussion of habituation, should also be cited.

2) Abstract, sixth line, should read: "Cyclase-dependent".

3) Introduction section, fourth paragraph: Substitute "4 minute-long continuous odors".

4) Subsection “The decrement in osmotactic response conforms to habituation parameters”, third paragraph: Figures are mistakenly referenced; they should read 3C, 3D, and 3E.

5) Subsection “Inhibitory local interneurons are necessary for refractory period maintenance”, second paragraph: It should be briefly stated what the effect of expressing the temperature sensitive Shibire mutation is for those readers who are the not *Drosophila* geneticists. In addition, the reference to "15-22…and 30-32…GABAergic, inhibitory local interneurons" should be explained.

6) Subsection “Inhibitory local interneurons are necessary for refractory period maintenance”, fourth paragraph: The meaning of the second sentence will be completely obscure to most readers.

7) Subsection “The MBs are essential for refractory period maintenance and dishabituation”, third paragraph, start of eleventh line: Rewrite as "not a consequence".

8) Legend, Figure 1: It should be explicitly stated that the odor concentrations are 1X.

9) Figures 1A,C and 2A,B,C: The x-axis of the graphs in Figure 1 represents time of exposure (to an odor), whereas the x-axis of the graphs in Figure 2, which is labeled identically, represents time after exposure (to 4 min of an odor). This is confusing. The x-axes for the graphs should be given different labels.

10) The phenomenon shown in Figure 2A and 2B is formally known as spontaneous recovery. This should be stated in the figure legend.

11) Legend, Figure 4. Throughout this figure, the phrase, "decreased OCT avoidance" is used to describe the resulting behaviors. While this is accurate, it would be much more informative and meaningful to describe the results as showing "increased [or enhanced or facilitated] habituation".

12) Legend, Figure 5A: Instead of describing the flies as having shown "significantly reduced OCT avoidance", why not say that they exhibited "significantly enhanced habituation to OCT"? Also, the phrases "decreased/reduced OCT avoidance", "premature devaluation of OCT avoidance", and "attenuated OCT response" should be rephrased in terms of the effect on habituation. (See #11 above.)

---

## [Author Response]

The paper will be improved overall by including a more complete discussion of how the different phases and mechanisms of habituation operate in concert (even if this involves some speculation) integrating these findings into the existing literature on habituation, not only in Drosophila, but also, where relevant, in other organisms. In addition the paper could be overall better communicated by carefully choosing terms most appropriate to the message of the paper, and also are consistent with previous usage. The Abstract, Introduction and Discussion would benefit from a deep re-write, to accommodate revisions as well as to clarify that the new findings pertain to (very) short term habituation and not to longer-term forms of this process.The paper will also be improved by addressing each of the comments detailed in the individual reviews below. For example, by considering the involvement of sensory adaptation in their 1-, 4-, and 30-min exposure protocols; and the potential effects of repeated but briefer exposures (e.g. 4 x 1-min vs 1 x 4-min).The reviewer discussion also noted the need to explain the observation that silencing GH146 neurons does not affect the behavior of naïve flies to OCT (Figure 4D), which is potentially confusing given that OCT is sensed at least in large part by GH146 expressing neurons.

This is an excellent point and one we wondered about ourselves. We present what we think is the most parsimonious explanation in the second paragraph of subsection “Excitatory projection neurons are essential for habituation latency”. We have published this before (Acevedo et al.,2007) and we do see this when silencing these neurons with Shibire (Figure 4D) and Tetanus Toxin (Figure S1 E). We hypothesize that the signal utilizes other neuronal routes to produce the observed avoidance behaviour. One such route may be the oACT, or yet undescribed neurons not marked with the GH146 driver. In addition, electrical synapses not affected by our silencing methods, could mediate the observed avoidance behaviour.

Separate reviews (please respond to each point):

Reviewer #1:

The manuscript reports a novel method to evaluate habituation to odorant stimuli in Drosophila. In essence, the new method uses a 4 min odorant pre-exposure in contrast to the 30 min protocol used in other labs. Genetic manipulations to silence or excite specific subsets of neurons in the olfactory pathway are used here to identify groups of cells involved in the different phases of the 4 min habituation. The results clearly point towards a diversity of mechanisms which render habituation as a rather heterogeneous learning process. This is an idea shared by many colleagues in the field which makes the manuscript welcome.There are, however, some issues that need to be addressed, in particular when drawing functional interpretations and circuits:1) In addition to chemical synapses, neurons of the olfactory pathway are connected by electrical coupling. Actually, this coupling is stronger than the chemical communication between eLN and PN according to studies previously published (albeit not quoted) (Yaksi et al., 2010; Huang et al., 2010). Electrical coupling is also functional, although weaker than the chemical communication, between eLN and iLN, and among PN neurons. In the experiments in which tetanous toxin or shibire are used to silence specific groups of neurons, the electrical synapses remain unaffected, hence communication between the referred groups of neurons must be still operational. This condition surely will affect the interpretation of the results summarized in Figure 8.

We agree with the reviewer and we are thankful for pointing out this omission in a possible interpretation of the results. To rectify this, we added discussion of the potential effects of the excitatory eLNs on projection neurons and inhibitory LNs through gap junctions (subsection “Inhibitory local interneurons are necessary for habituation latency”, paragraph four). In addition, a section was added in the discussion of the results of iPN inhibition (subsection “Excitatory projection neurons are essential for habituation latency” paragraph four), since the iPNs were also found connected to ePNs through chemical synapses (Shimizu, et al., 2017). We have added the electrical couplings detailed above in the model on Figure 8.

2) The conclusion that the 30 min and 4 min habituations are mechanistically (Rut dependence) and spatially (LH versus AL) different processes is attractive, but, given the large difference in the times involved, it appears counterintuitive. How could a 30 min process be kept constrained within the AL?

As the reviewer points out, our collective data strongly suggest different mechanisms for 4-minute and 30-minute habituation to OCT. Habituation to 4 minutes of OCT engages synapses between inhibitory projection neurons and their targets in the LH. In contrast, habituation to the prolonged 30-minute OCT exposure is consequent of the inhibition of excitatory projection neurons by local interneurons in the antennal lobe, resulting in decreased ePN output (Das, Sadanandappa et al., 2011). Therefore, habituation to 30 minutes of continuous odor stimulation is mediated by the antennal lobe, inasmuch as it decreases responsiveness to the odor stimulus, but is not constrained there *per se*, because it has to be expressed via centers that modulate the behavioral output measured as OCT avoidance decrease. Similarly, habituation latency is mediated by the mushroom bodies, but does not reside there *per se*. It is possible then that the lateral horn (LH) plays a role in habituation to 30-minute odor stimulation because the potentiated inhibition of projection neurons would decrease their output to the LH, potentially modifying the behavioral output. Hence, habituation to 4-minute and 30-minute stimulation could rely on decreased LH activation, but the mechanisms used to this end are different. For habituation to 4-minute stimulation the decreased response to the odor is attributed to synaptic modulation of iPNs, ePNs and neurons of the LH, while habituation to 30-minute stimulation is driven by the antennal lobe iLNs to decrease projection neuron activation at the antennal lobe. The differences between these two types of habituation are detailed in the Discussion section (subsection Habituation after prolonged odor stimulation” third and fourth paragraph).

3) The use of concentrated and diluted odorant solutions is considered as "strong" and "weak" stimuli here and throughout the literature. Being strict, however, we cannot use these terms in the absence of electrophysiological evidence showing higher frequency firing in the first case with respect to the second. If the key neurons involved would be of the low responder type (firing sparse action potentials only), a diluted odorant stimulus would elicit the same response as a concentrated one. At least the MB neurons belong to the sparse responder type.

We agree with the reviewer and we have added a section in the results pointing out that the firing frequency of the implicated neurons after exposure to different Octanol concentrations is not known (subsection “The decrement in osmotactic response conforms to habituation parameters” third paragraph). However, imaging experiments on glomerulus activation during 3-Octanol exposure showed that the glomeruli are responding in a concentration dependent-manner, with high concentration doubling up their response (Yu, et al. 2004). Even though this comparison was made using different odor concentrations than those used in our study, it nevertheless suggests that OCT-responding neurons are activated in a concentration-depended manner and they appear not to fire sparsely, apparently then independently of stimulus strength. Furthermore, if the different concentrations would not affect the neuronal activation, we would expect a similar habituation “rate” between 1X and 0.1X Octanol, which is not what we observe.

4) The use of Gal4 drivers is quite extended in Drosophila neurobiology and the specificity of their domains of expression is strongly emphasized. Detailed scrutiny of these expression domains, however, shows far more variability than usually admitted. For instance, based on the GTRACE technique most Gal4 drivers exhibit variable expression during development and among individuals (Evans et al., 2009). One driver used here, GH298-Gal4, has been reported to exhibit transient expression in neurons of the adult brain (Casas-Tinto et al., 2017).

We have tried to overcome potential non-specific expression as that observed in some Gal4 lines, using more than one line to drive expression in a distinct neuronal subset, when this was possible. GH298*Gal4, a line we used to express the UASshibire^ts^*transgene in the local olfactory interneurons also marks other clusters of cells along the brain (Casas-Tinto et al., 2017). For this reason, GH298 was not the only driver used to silence the inhibitory local interneurons, but also complemented by LN1Gal4 that also marks inhibitory local interneurons. This decreases the probability, but does not eliminate potential individual expression level variability that could impact the results.

All these comments do not diminish the value and general interest of the manuscript but represent issues that should be considered in the interpretation of the data.Finally, a few minor points on typos:Materials and methods section, third from last paragraph: 3econds >> 3 secondsSubsection “Hydroxyurea treatment”: extend >> extentDiscussion paragraph nine: are consistent the >> consistent with

All minor comments of the reviewer have been addressed, and we are obliged by his attentive reviewing.

Reviewer #2:

This paper reports a behavioral protocol to study decrement in responses to aversive or attractive odors following prior exposure to the same or a different odor. The results are novel and potentially interesting to a broad readership. However, before the paper is accepted for publication, several fundamentally important issues must be resolved or carefully qualified to avoid confusion in the literature.1) Sensory adaptation and olfactory habituation. The olfactory behaviors described here do conform with some of the criteria of Habituation, including response decrement following prior exposure, spontaneous recovery, and dishabituation (cf. Rankings et al., 2009, Neurobiol. Learn Mem, not cited). However, a commonly adopted protocol of repetitive stimulus delivery is missing here.Instead of the repetitive nature of stimulation in other paradigms, relatively prolonged, continuous, exposures to the odors OCT, ETA or BNZ were adopted. While the fly would take only fractions of a second or seconds to respond, it endures continuous, passive exposure to the odors for minutes in this protocol. It makes one wonder how much sensory adaptation takes place during the prolonged 1-, 4-, or even 30-min exposure? Earlier research from Siddiqi's and others' times needs to be carefully reviewed. How do the OSN spikes or even EAG recordings change over such time scales, i.e. how much receptor adaptation actually occurred during minutes of exposure?

To address the reviewer’s useful suggestion, we performed olfactory habituation experiments using repetitive pulses of odor stimulation. As we explain in the Results (second paragraph) and the Materials and methods sections (subsection “Olfactory habituation” second paragraph), we divided the 1 minute of continuous odor exposure into two 30-second odor pulses with 8 seconds interstimulus interval, the 4-minute odor exposure into four 1-minute odor pulses with 15 seconds interstimulus interval and finally the 30-minute exposure was divided into three 10-minute pulses with 2.5 minutes interstimulus interval (Figure 7—figure supplement 1A, B). The length of the interstimulus interval was kept at a quarter of the odor exposure duration, since adaptation has been correlated with stimulus duration.

*Drosophila* OSNs reportedly adapt their responses after odor pulses even as brief as less than 30 seconds (Cao et al., 2016). This suggests that adaptation to olfactory stimuli potentially affects all behavioral protocols in *Drosophila* utilizing odor stimulation lasting 30 seconds or more. Adaptation adjusts odor sensitivity dynamically, allowing both desensitization and prevention of saturation (Cao et al., 2016) and it is thought to adjust receptor activation when the background odor is fluctuating (Nagel and Wilson 2011). In this study, we focus on responses of second- and higher-order neurons after continuous exposure to the odor. Since olfactory adaptation occurs quickly, its effect on OSN activation will be omnipresent irrespective of stimulus duration. However, our data indicate that 1-minute of OCT exposure does not affect subsequent responses to the same stimulus in wild type flies, strongly suggesting that it is not the OSNs where the response is attenuated due to adaptation, but neurons downstream in the olfactory pathway. Moreover, if the observed osmotactic decline was a consequence of OSN adaptation, increased odor concentration would lead to faster adaptation and thus, faster response attenuation, which we do not see. In contrast, increased concentration leads to slower habituation, in agreement with the characteristics of habituation (Thompson and Spencer 1966, Rankin et al., 2009). The sensory adaptation question has been addressed in the Results section (subsection “The decrement in osmotactic response conforms to habituation parameters” fourth paragraph).

2) The authors have not made a convincing case that an optimal and relevant protocol has been worked out through a comprehensive, systematic experimental trials. The current approach has a relatively poor temporal resolution. Empirically, varying or modifying some simple, basic experimental conditions will produce new, useful information. For example, how do the outcomes differ for the prior exposure schemes of 2X30sec vs 1X1min? 4X1min vs 1X4min? or 3X10min vs 1X30min (Figures 7G and Figure 7—figure supplement 1D-F)?In other words, is the cumulative exposure time or repetition exposures more effective?

Systematic exposure to one to four minutes of continuous odor demonstrated that a minimum of three minutes of exposure is required for sufficient response attenuation (Figure 1). We have used shorter exposures which also do not produce attenuated osmotaxis, but we do not include that data because of our focus on the 1 minute mark. However, we addressed the reviewer’s question regarding cumulative exposure time versus number of repetitions. The results (Results section paragraph two and subsection “The Rutabaga Adenylyl Cyclase is essential for normal olfactory habituation” third paragraph) show no significant difference in response attenuation between continuous and repetitive stimuli. This suggests that the total duration of the exposure and not the number of repetitions of odor pulses is significant for this form of habituation (Figure 1—figure supplement 1A, B, Figure 7—figure supplement 1A).

3) In Figures 7G and Figure 7—figure supplement 1D-F, rutabaga mutants did not habituate after 30 minutes OCT exposure whereas WT displayed clear habituation but its "dishabituation" required "yeast puff". The results are intriguing but should be presented in the context of distinguishing different forms of sensitization. Why does electric shock or mechanical vortexing no longer work? What would be the time course of spontaneous recovery for this form of long-term "olfactory habituation"?

We are assuming the reviewer meant different forms of habituation. We thank the reviewer for raising a very important distinguishing point between the two olfactory habituation paradigms. Indeed, the experimental evidence indicates that habituation after 4 and 30-minutes of odor exposure are different processes that use distinct neuronal subsets and apparently molecular pathways to drive response attenuation. The effect of different stimuli as dishabituators after habituation to 4- and 30-minute odor exposure lends further credence to this hypothesis. The inability of mechanical stimuli to reverse habituation after 30-minutes of odor exposure suggests that these stimuli are unable to reverse or reset the proposed inhibition of projection neurons by the local interneurons in the antennal lobe. A puff of yeast odor however, likely engages additional or distinct neurons as it predicts food and may therefore act as a sensitizing stimulus. These results suggest that distinct dishabituators likely recruit different neuronal subsets to modulate the habituated response. Such neuronal circuits and the effect of different dishabituators in recovery of the response are currently under investigation. The effect of different dishabituators after 4 and 30 minutes of odor exposure is discussed in paragraph two of subsection “Habituation after prolonged odor stimulation”.

Regarding the question of spontaneous recovery upon habituation to 30 minutes of odor exposure. This has been shown to occur upon 30 minutes of rest (Das, Sadanandappa et al., 2011), but shorter rest periods had not been tested. We now provide evidence that spontaneous recovery to 30 minutes OCT habituation occurs even after 6 minutes of rest, as for habituation to 4 minutes of OCT. Our experiments also confirmed (paragraph three subsection “The Rutabaga Adenylyl Cyclase is essential for normal olfactory habituation”) spontaneous recovery after 30 minutes of rest (Figure 7—figure supplement 1C) and demonstrated that even 6 minutes of rest are sufficient for spontaneous recovery of the response after 30 minutes of odor exposure (Figure 7—figure supplement 1B).

4) The authors propose that there are two phases of their habituation process, which could be dissected in genetic manipulations and be associated with different identified neural circuits. The proposed concepts of "refractory period maintenance" and "habituation onset" appear to be mostly based on the differences between 1min vs 4 min exposure and the rut/AC mutantational effects. Is this truly the only interpretation? Could it be two separate kinds of sensory processing with kinetics over different time scales/ temporal domains?

We basically agree with the notion of two types of sensory processing with distinct time scales. As we propose in the Discussion section, latency and habituation appear to engage distinct sensory processing circuits. The 1-minute odor exposure recruits antennal lobe local interneurons, excitatory projection neurons and mushroom bodies to maintain the avoidance response (paragraph one, subsection “Olfactory habituation phases are mediated by distinct neuronal subsets”), while continuous activation of inhibitory projection neurons by longer odor exposure is inextricably linked to attenuation of the response (paragraph four). The kinetics of inhibitory projection neuron activation and their output on downstream neurons is likely used by the brain as a marker of exposure duration, signaling to inhibit odor avoidance after odor exposure sufficient to devalue the stimulus (paragraph three, subsection “Olfactory habituation phases are mediated by distinct neuronal subsets”).

5) The authors may want to review and compare their results with the previous Drosophila habituation paradigms for aspects of repeated stimuli, spontaneous recovery, and dishabituation. A more global picture may emerge to enhance the presentation and strengthen their contributions to the field. A partial list:i) The giant fiber-mediated jump & flight escape/startle reflex (repetitive light flash / air puff dishabituation)ii) Proboscis extension reflex (also in honey bees).iii) Olfactory startle jump reflex (repetitive odor puffs / vortex dishabituation)iv) Landing response (repetitive visual looming / vibration dishabituation)In particular, habituation process of the escape reflex following startling stimuli and other processes also involves dunce and rutabaga. These lines of observations may converge upon a short time scale process that is related to the relevant sensory information processing.

As the reviewer suggests, we added a section in Discussion, comparing our paradigm with other habituation paradigms in *Drosophila*, comparing and contrasting relevant parameters (paragraph one). However, we focused on related paradigms and habituation parameters pertinent to our own work presented herein. We do agree, that a thorough review and comparison of all these paradigms is in order probably as a stand-alone review, but we do not believe that this is within the remit of this manuscript.

Reviewer #3:

This is an interesting paper on the neural mechanisms underlying a novel form of short-term olfactory habituation in Drosophila. The experiments have been carefully performed and the data appear reliable. However, in my opinion the paper is flawed by the terminology used to describe the basic behavioral phenomena; the terms are poorly chosen and make the findings unnecessarily difficult for the reader to understand. Below are my recommendations for improving the paper.Major comments:1) The paper identifies and analyzes two basic aspects of the olfactory habituation: the minimum time of exposure to an odorant of a given concentration sufficient to trigger habituation, which is termed the "refractory period"; and the decrease in the flies' responsiveness to an odorant, which is termed "habituation onset".First, refractory period is a poor choice because it connotes two discrete exposures to a stimulus, for example the refractory period between the ability of two depolarizing current pulses to evoke an action potential in a neuron. By contrast, the odorant exposure used to determine the minimum exposure time required to elicit habituation is continuous. This phenomenon should be given a more accurate name, such as "critical time of stimulus exposure" or "minimum (stimulus) exposure time sufficient for habituation induction".Second, what the authors mean by "habituation onset" is simply whether or not the phenomenon of habituation is present; they are not referring specifically to the onset of habituation, but, rather, to habituation itself. The reference to "onset" is confusing. The authors should simply use the word "habituation" or, where appropriate, "induction of habituation".

We thank the reviewer for these highly relevant nomenclature suggestions, which we have adapted and have modified the text accordingly. The terms “refractory period” and “habituation onset” have been replaced by “habituation latency” and “habituation”, respectively.

2) It is claimed that the form of habituation reported in the paper lacks the stimulus generalization that characterizes many (most?) forms of habituation, as originally noted by Thompson and Spencer. But the basis for the claim here that the habituation does not generalize is a comparison between two odors of the same (negative) valence and two odors of opposite valence. These are hardly exhaustive tests for the presence of stimulus generalization; the authors should modulate their claim accordingly.

We are wondering as well. To examine whether the attenuated response is generalized between odors, we pre-exposed the flies to an aversive odor (OCT) and tested them against another aversive odor (BNZ) (Figure 2C) or pre-exposed them to an attractive odor (ETA) and tested them against an aversive one OCT (Figure 2—figure supplement 1). Since we have used only two dissimilar odors to study generalization, the reviewer’s point is right, and we have noted the possibility of generalization to similar odors in the respective Results section (subsection “The decrement in osmotactic response conforms to habituation parameters”). To make a conclusive statement about this, a lot more odor pairs need to be tested.

3) Initially, habituation to attractive odorants, ethyl acetate (ETA) and 2,3-butanedione (BUT) is examined; however, after the experiments presented in Figure 1 are summarized, habituation to attractive odorants is not discussed further. Therefore, the extent the cellular models presented in Figure 8 pertain to attractive, as well as repellant, odorants is unclear. This point should be discussed.

The reviewer is right, and we fully agree that it is unclear whether the proposed model could work for attractive odors as well. We have added a section on discussing the possibilities for attractive odor habituation in the Discussion (subsection “Olfactory habituation phases are mediated by distinct neuronal subsets”).

Minor Comments:

All the minor comments of the reviewer have been addressed. We thank the reviewer for the detailed and thoughtful comments.